**Investigation**

# Detecting and quantifying networks of biological kinship via exponential family random graph models

Adam B. Rohrlach [ID] ,[1,2,*,‡] Guido Alberto Gnecchi-Ruscone,[3,4] Zuzana Hofmanová,[1,5] Zsófia Rácz,[6] Matthew Roughan,[7] Wolfgang Haak,[1] Jonathan Tuke[7,‡]

[1]Department of Archaeogenetics, Max Planck Institute for Evolutionary Anthropology, Deutscher Platz 6, Leipzig 04103, Germany
[2]School of Biological Sciences, University of Adelaide, North Terrace campus, Adelaide 5005, Australia
[3]Archaeo- and Palaeogenetics, Institute for Archaeological Sciences, Department of Geosciences, University of Tübingen, Geschwister-Scholl-Platz, Tübingen 72074, Germany
[4]Senckenberg Centre for Human Evolution and Palaeoenvironment at the University of Tübingen, Geschwister-Scholl-Platz, Tübingen 72074, Germany
[5]Department of Archaeology and Museology, Masaryk University, Žerotínovo nám. 617/9, Brno 601 77, Czechia
[6]Institute of Archaeological Sciences, ELTE - Eötvös Loránd University, Múzeum krt. 4/B, Budapest 1088, Hungary
[7]School of Mathematical and Computer Sciences, University of Adelaide, North Terrace campus, Adelaide 5005, Australia

*Corresponding author: Department of Archaeogenetics, Max Planck Institute for Evolutionary Anthropology, Deutscher Platz 6, Leipzig 04103, Germany. Email: adam_ben_rohrlach@eva.mpg.de.
‡These authors contributed equally to this work.

Genetic relatedness between ancient humans can help to identify close and distant connections between groups and populations, uncovering signatures of demographic histories such as identifying mating networks or long-range migration. Critical to researchers are the characteristics that connected individuals, or groups of individuals, share, and how these characteristics interact and are correlated. Here we use Exponential Random Graph models as a method to explore demographic and contextual parameters that may help to explain the significant drivers of the topology of mating networks, as well as to quantify their effects. We show through simulations that model selection and coefficient estimators facilitate the exploration of such networks, and apply the method to individuals from a collection of Avar-associated cemeteries from the Carpathian Basin dating to the 6th to the 9th centuries CE.

Keywords: Genetic relatedness; Kinship; IBD

## Introduction

Past and present human populations actively and passively engaged in social, commercial and cultural practices that led to the formation of mating networks (Held et al. 2016; Romano et al. 2020; Ning et al. 2021; Csardi et al. 2023; Pearson et al. 2023; Wang et al. 2025b). These networks, which can be thought of as massive family pedigrees, were distributed across time and space. The processes that led to the formation of these networks, and which societal norms mediated their shape and topology, are largely unknown, and it is unclear how to leverage the evidence in the observable archaeological record to fill this gap.

Recent developments in ancient DNA (aDNA) analyses and ever-growing sample sizes (both per site and in the field of archaeogenetics overall) have allowed researchers to estimate snapshots of these networks of close and distant relatedness from both low- and high-coverage sequence data (Popli et al. 2023; Ringbauer et al. 2023; Rohrlach et al. 2025; Alaçamlı et al. 2024). These methods have been applied to whole-cemetery analyses of single sites, as well as intra- and cross-regional studies, to reconstruct deep pedigrees (Fowler et al. 2022; Villalba-Mouco et al. 2022; Rivollat et al. 2023; Gnecchi-Ruscone et al. 2024; Penske et al. 2024; Wang et al. 2025a). However, these methods can also uncover inter-regional connectedness due to contact, trade and exchange, individual mobility, and migration.

Archaeogenetic studies of relatedness networks have mostly focused on single network statistics calculated without taking into account potential network-wide dependency structures, and answer only limited hypotheses about single parameters, without quantifying these effects or considering potentially interacting variables (Rivollat et al. 2023; Gnecchi-Ruscone et al. 2024). Other studies have applied clustering methods to explore embedded clusters of individuals to explore drivers of this clustering (Ferrando-Bernal et al. 2020; Gretzinger et al. 2025; Szécsényi-Nagy et al. 2025).

Exponential Random Graph models (ERGMs) are a powerful tool that have been used to analyze both sparse and dense networks with interacting covariate effects and a wide range of complex dependencies (Schweinberger et al. 2020). ERGMs have been used to explain the structure of social networks, owing to their ability to detect significant predictors even when the overall model explains only a relatively low percentage of the variability in the network, such as might be expected in mating networks (Sharifnia and Saghaei 2022). Network analyses, and specifcally ERGMs, have also been used to analyze networks defined by archaeolgical data, making them a natural and familiar tool for analyzing ancient DNA (Brughmans et al. 2014; Amati et al. 2020; Wang and Marwick 2021; Amati 2023; Mazzucato et al. 2025). Hence, ERGMs are a sensible tool to apply to networks constructed from genetic relatedness, but that also contain genetic, archaeological

and anthropological variables measured on the individuals. Importantly, using ERGMs, we can quantify the impact of variables on the connectedness of individuals, and yield measures of fold-increase in the probability of "types" of individuals sharing a genetic relationship.

To highlight the usefulness of ERGMs to efficiently detect variables of importance and to quantify their impact, we use simulated networks and Bayesian Information Criterion based model selection to show that we can reliably recover the correct, and hence best-fitting, models. We then explore how sample size and the relative effect size of variables impact the performance of ERGMs to give researchers a baseline for when ERGMs can be reliably applied. Finally, we use ERGMs to analyze an empirical data set from four Avar-associated cemeteries from the Carpathian Basin to show how ERGMs can be used to combine archaeogenetic and archaeological data into one cohesive analysis.

## Materials and methods
### The statistical model

Consider an undirected network $G$ with a set of nodes $V(G) = \{1, 2, \ldots, n\}$ where $i$ represents the $i^{\text{th}}$ individual (with $|V| = n$ total individuals in the study) and a set of edges $E(G)$ which represent pairs of individuals with a given type of relationship, such as sufficient shared IBD (identical by descent).

It is common to form such a network starting with a measure $e_{ij}$ of the connectedness of individuals $i$ and $j$ formed from genetic or other data, where 0 indicates no measured connection, and a high value indicates a strong relationship. We then construct the network through thresholding, i.e. we define the network adjacency matrix $\mathbf{Y} = [Y_{ij}]$, where

$$Y_{ij} = \begin{cases} 1, & \text{if } e_{ij} \geq c, \\ 0, & \text{if } e_{ij} < c, \end{cases} \tag{1}$$

where $c$ is a cutoff that determines a significant connection. An edge $(i, j)$ exists in $E(G)$ if and only if $Y_{ij} = 1$, and, in the case of IBD networks, the choice of $c$ reflects the research questions of interest.

The choice of cutoff might be determined through a statistical model of the data-measurement process, by calibration to some "ground-truth" data, for instance, obtained through *in silico* simulations or tested through sensitivity analysis (Supplementary Section S5.B and Supplementary Fig. S8 and S13). We then treat our network as a random sub-sample of the complete network, the sub-sample being determined via measurement noise, and the presence or absence of measured individuals.

Additionally, variables are recorded for individuals and edges, denoted $\mathbf{V}$ and $\mathbf{W}$, respectively. For example, node variables $\mathbf{V}$ record demographic parameters about each individual such as the genetic sex, age at death or, burial context. The edge variables $\mathbf{W}$ record pairwise shared measures, such as the distance between burials or the difference in mean $^{14}$C date.

Of interest is the probability of an edge existing in the network, denoted

$$p_{ij} = P\left(Y_{ij} = 1 \mid \mathbf{V}, \mathbf{W}\right), \tag{2}$$

conditional on these variables. Moreover, we assume locality in the sense that the probability of connection between individuals $i$ and $j$ depends only on the variables of these individuals, i.e. $\mathbf{v}_i$, $\mathbf{v}_j$ and $\mathbf{w}_{ij}$. We can further construct synthetic edge variables $\mathbf{z}_{ij}$

from pairs of node attributes, e.g. we might consider *homophily* measures such as $z_{ij} = I(v_i = v_j)$, where $I(\cdot)$ is the indicator function (which takes the value 1, when its argument is true, and 0 when false).

As in logistic regression, we work in the log-odds space and instead look to make inferences about $\log(p_{ij}/(1 - p_{ij}))$. We perform a network regression to model the log-odds of any two individuals being connected given the measured variables, i.e.

$$\log\left(\frac{p_{ij}}{1 - p_{ij}}\right) = \theta_0 + \sum_{k=1}^{K} \theta_k z_{ijk} + \sum_{\ell=1}^{L} \lambda_\ell w_{ij\ell}, \tag{3}$$

where $\theta_k$ and $\lambda_\ell$ are the coefficients for the $K$ node and $L$ edge variables $z_{ijk}$ and $w_{ij\ell}$, respectively (Supplementary Section S1). For simplicity we restrict the following examples to node attributes, and hence assume $L = 0$. It must be noted that this allows us to consider only dyad-independent ERGMs, and we make this choice for the reasons of computational efficiency, due to potentially large numbers of nodes in the network, and model degeneracy due to the likely relative sparsity of the network (Karwa et al. 2022). However, we show under simulations that this does not significantly effect the performance of the method (Supplementary Section S3.A).

While this approach has many similarities to logistic regression, there are some critical differences. First, the assumption of independence for observations: in the network case, connectedness of nodes can not be assumed to be independent. To see this, consider that if individuals $i$ and $j$, and individuals $i$ and $k$ are closely genetically related, then it is likely that individuals $j$ and $k$ are also closely related.

Second, the nature by which nodes manifest relationships can be varied—consider the example of individuals buried at three sites: A, B and C, with genetic sexes "Sex 1" and "Sex 2" (for a full discussion see Supplementary Section S2 and Supplementary Figs. S1–S7).

It might be that simple homophily (nodematch) is the primary driver determining connectedness, that is, a pair of nodes that *share* an attribute are more likely to be connected. An example of this may be that individuals who share the same burial site are more likely to have been genetically related.

Simple homophily requires that the increase in connectedness probability is the same for each site. An alternative is differential homophily (differential sitematch), i.e. connectedness is still more likely for individuals that share the same attribute, but the value of the attribute (that they share) is also important. For example, imagine that site A was made up of one pedigree, but sites B and C are made up of several, unrelated pedigrees. It may still be the case that individuals buried at the same site are more likely to share a genetic connection, but that since sampling site A will always produce pairs from the same pedigree, that site A was more interconnected than sites B or C. Hence, two individuals buried at site A are more likely to be connected than two individuals buried at site B (or C).

Finally, we consider the idea of attribute mixing (nodemix). In this case, it no longer matters whether the attributes match, but the combination of the two attributes is of importance. An example of this could be that (as in the previous example) site A is more interconnected, but sites B and C are close to each other. Hence, while individuals within sites are quite likely to be related, and more so for site A, individuals between sites B and C are also more likely to share a genetic connection than between A and B (or A and C).

From a statistical modeling perspective, these are increasingly complex models, i.e. they require additional parameters to be

estimated. See Supplementary Section S2 for a list of the models, but for the moment note that in our examples the first requires one parameter (the degree of homophily), the second requires three (one homophily parameter for each region) and the third requires up to six parameters to consider all pairs of regions.

## Model selection

Model selection is the process of choosing which variables are important in determining the model for connection probabilities, i.e. which $z_{ijk}$ terms are best included in the model. A typical problem is that more complicated (more highly parameterized) models will often fit the data more exactly, but without generalizability, i.e. without the ability to extrapolate to datasets other than the one measured. Such over-fitting is highly undesirable and usually avoided using one of several *Information Criteria*.

Here we use the Bayesian Information Criterion (BIC) (Schwarz 1978). BIC prevents over-fitting by introducing a penalty term for additional model parameters. We use BIC, as opposed to alternatives such as Akaike's Information Criteria, because (i) it is more conservative, and (ii) it is more naturally matched to the type of models being used here (Drton and Plummer 2017). Specifically, BIC is preferred when the sample size is much greater than the number of parameters in a model, which is the case for even relatively small networks where the number of potential edges grows factorially with the number nodes. We also calculated the corrected AIC (AICc) (Sugiura 1978), but found that due to large number of observed pairs of nodes relative to the number of parameter values, that it performed identically to the AIC (Supplementary Fig. S12). We also compare the performance of BIC to AIC and a P-value-based approach to show that BIC outperforms all other model selections methods.

## Simulation performance and power analyses

To test the performance of ERGMs to be able to identify the correct model we simulated 100 realisations of each model (see Supplementary Section S2 and Supplementary Tables S1–S2), yielding a total of 700 simulations. We then applied the BIC model selection criteria, and recorded the best fitting model for each simulation. From this, we are able to calculate a range of performance statistics, including accuracy and specificity.

We were also interested in calculating a lower cut-off for how impactful variables must be on the network to be detected, dependent on the number of edges in the network. We simulated networks from the site match model, with the expected number of edges on a grid between ten and 10,000, and with values of $\theta_0 = -4$ and $\theta_1 \in \{0, 0.05, \ldots, 1\}$. We use a simple P-value cut-off of $\alpha = 0.05$ for coefficient significance. For each value of $|\mathbf{V}|$, we are then able to find the minimum value of $\theta_1$ such that the standard 80% power is achieved (Cohen 2016).

## Network clustering and centrality analyses

To investigate structure within the unweighted Avar network we performed Louvain clustering on a network with edges weighted by the total amount of shared IBD, and calculated degree and betweenness centrality using the *igraph* package (Csardi and Nepusz 2006; Antonov et al. 2023).

## Results

### Performance of ERGMs on simulated data

To evaluate the accuracy of model selection, we simulated 100 realisations for each of the seven possible models on networks with $n = 300$ nodes, and parameters values as described in Supplementary Section C. For each realisation we used the minimum BIC value to assign the best possible model. Interestingly, no single realisation was incorrectly assigned to the true model (see Fig. 1). We also found that BIC outperformed using AIC or the P-values associated with the coefficients of the model for model selection (see Section Supplementary Section S4 and Supplementary Figs. S9–S11).

Following this, we were interested in assessing the statistical power of ERGMs when performing model selection using BIC. To start with, we simulated realisations of the site match model (see Supplementary Section S2.B) with varying values of $\theta_1$. We let $\theta_1$ take values from zero to one, in steps of 0.05, which equates to a fold-increase in probability of between 1 and 2.5 (see Fig. 2). We find that when $\theta_1 = 0$, that is when there is no fold-change, power is the expected 5%, matching the significance level of $\alpha = 0.05$. We also find that for $\theta_1 \geq 0.35$ (approximately 1.41-fold increase), power is 100%. Finally, we find that we achieve the standard 80% for $\theta_1 \geq 0.25$, which is equivalent to a fold-increase in probability of only 1.27, meaning that even subtle variables can be detected using BIC.

We tested whether the power to detect these variables may be affected by uneven sampling. To assess this, we repeated the above experiment, but this time simulated that 25% of nodes are "Sex 1", and that the remaining 75% are "Sex 2". We find that power is slightly decreased in the unequal case for smaller values of $\theta_1$ (see Fig. 3). However, 80% is still achieved relatively quickly at $\theta_1 = 0.3$ (approximately 1.34-fold increase), and that 100% power is achieved quickly after this.

Finally, we were interested in how the number of edges in the network affected the power of BIC to correctly perform model selection. As it is technically the number of edges in the network (which is stochastic) will influence predictive power, and not the number of edges, a derivation of the formula for the expected number of edges in a simulated network, given a number of nodes, can be found in Supplementary Section S4.C. For each number of expected edges, we returned the minimum value of $\theta_1$ that achieved 80% empirical power (see Fig. 4).

Unsurprisingly, we find that as the number of edges, increases, that the minimum fold-increase in probability decreases. This is expected, as more edges result in more information, and hence more statistical power. We also see that ERGMs are capable of reliably detecting even very subtle variables, with fold-increases of less than 1.5 for networks with just 500 edges (approximtely 150 nodes in this example), tending to a value of approximately 1.07-fold as the number of expected edges becomes large.

### Analysis of empirical data

Finally, we reanalyzed the networks presented in Gnecchi-Ruscone et al. (2024), where individuals share an edge if they share at least two blocks of IBD of length ≥12cM and at least one block of IBD of length ≥16cM , indicating a degree of relatedness of approximately at least seventh-degree. The networks represent individuals sampled from four nearby sites, associated with the early to late Avar period in Hungary (567–822 AD). Using measures of centrality, the authors found that the sub-networks connecting genetically male individuals were significantly more dense than either the sub-network connecting genetically female individuals, or the full network (Gnecchi-Ruscone et al. 2024). The network (see Fig. 5) consisted of 237 individuals, connected across sites, for which several variables were recorded including genetic sex, age (adult/subadult), body orientation, and whether they were buried with (a) a common item: an iron buckle, or (b) something rare and valuable: horse-riding equipment (or both). Note that

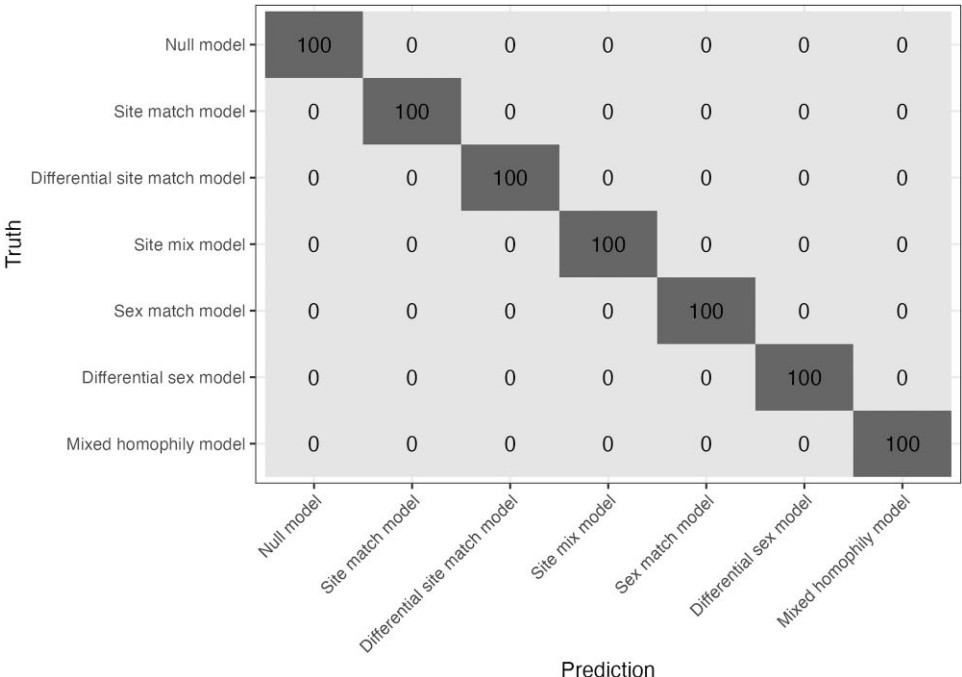

**Fig. 1.** A confusion matrix for the accuracy of model selection via BIC.

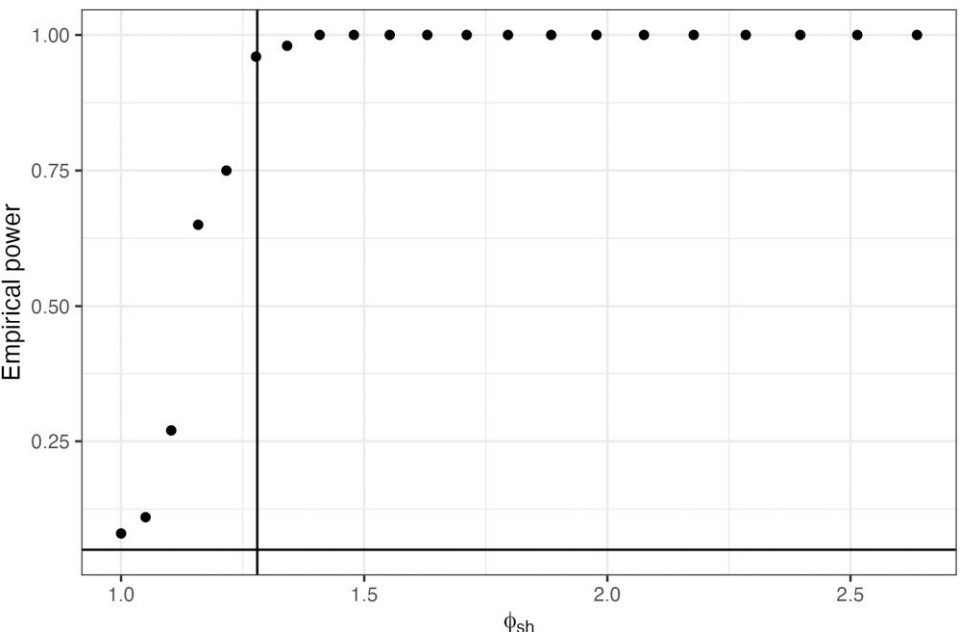

**Fig. 2.** Empirical power calculations for the sex match model. The x-axis indicates the fold-increase in probability if the sex matches for two nodes, and y-axis gives the value of the empirical power.

we performed a sensitivity analysis of the cut off $c$ to show that varying the minimum relatedness of two individuals did not change which variables were found to be significant (Supplementary Section S5.B).

Variables that we might expect would lead to higher probabilities of genetic relatedness (due to geographical and temporal closeness) are retained as significant: site and period (see Table 1). Since differential node match worked best for period, it is then the fact that individuals born during the Early Avar period and Middle/Late Avar period are more likely to be related to

individuals from the same time period (5.16- and 9.17-fold, respectively). Individuals buried at the same sites are also more likely to be related, and this differs from site to site as some sites are "more inter-related". Specifically, HNJ (549-fold), KFJ (857-fold) and KUP (691-fold) are more inter-related than RK (185-fold). This discrepancy in interconnectedness can also be seen when performing network clustering (so-called community detection) and inspecting the connectedness of the network using centrality measures (see Supplementary Section S5.C and Supplementary Fig. S14). The three visible clusters in RK can be separated into a

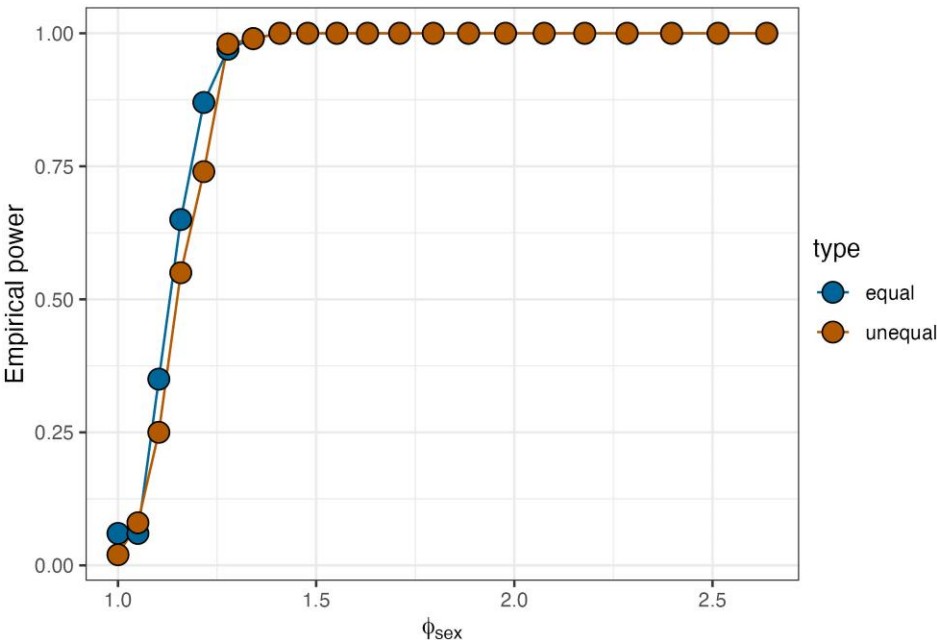

**Fig. 3.** Empirical power calculations for the sex match model. The x-axis indicates the fold-increase in probability if the sex matches for two nodes, and y-axis gives the value of the empirical power. Blue points indicate equal numbers of each sex (50% each), and orange indicates unequal numbers of each sex (25% and 75%).

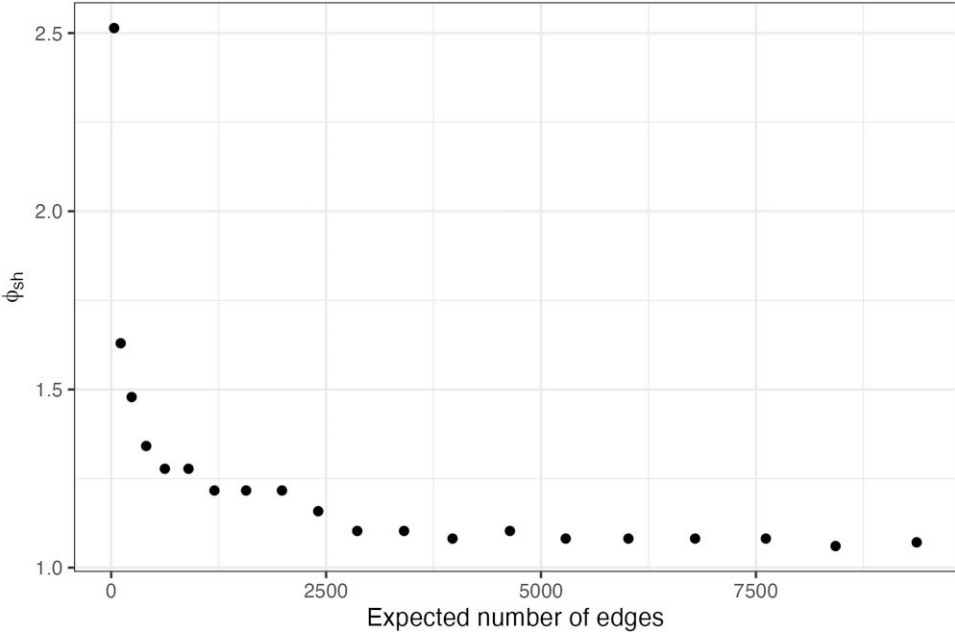

**Fig. 4.** Empirical power calculations for the sex match model. The x-axis indicates the expected number of edges in the network. The y-axis gives the minimum value of the fold-increase in probability if the sex matches for two nodes, that achieves an empirical power of 80%.

total six communities, indicating relatively low within-site connectivity, with these sites corresponding to the sub-pedigrees in Gnecchi-Ruscone et al. (2024). Similarly, KUP is split into two clusters. We observed that these two KUP clusters can be separated by period.

Interestingly, the age/sex of an individual was the only variable for which the more complex node-mix was selected (see Table 1). We observe that, compared to an adult male and an adult female, two adult males are far more likely to be related (6.62-fold) and

that two adult females are far less likely to be related (0.26-fold). This might indicate a social practice of patrilocality and/or female exogamy: where female individuals leave the place of their birth to find a mate and live near the family of their male partner. We also observe that the only other variables that lead to no change in probability compared to adult male/adult female pairs also involves adult females: adult female/ subadult female ($Z = -0.61$) and adult female/subadult male ($Z = 1.35$). This suggests that when female individuals left their place of birth to join a mate

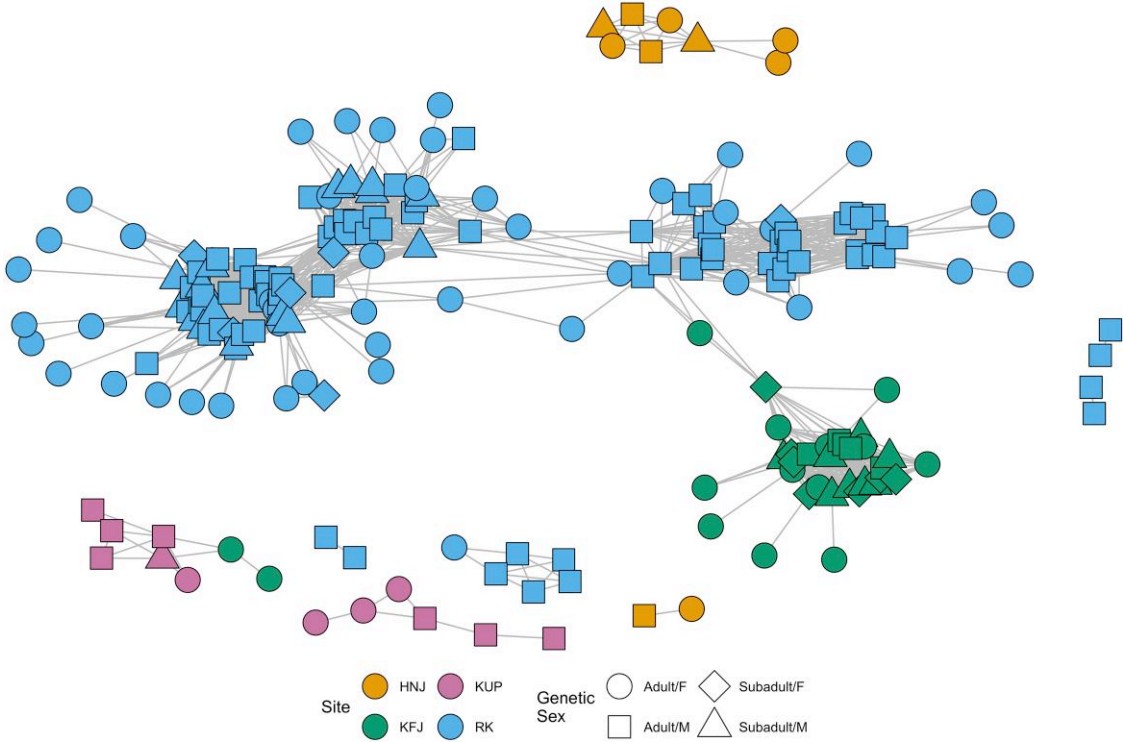

**Fig. 5.** The Avar network connecting individuals from Hajdúnánás-Fürj-halom-járás (HNJ, orange), Kunszállás-Fülöpjakab (KFJ, green), Kunpeszér-Felsőpeszéri út (KUP, purple) and Rákóczifalva Bagi-földek 8 (RK, blue). Shapes indicates the genetic sex and age of the individuals. Edges exist if two individuals share at least two blocks of IBD of length of at least 12cM, and one block of at least 16 cM.

**Table 1.** Summary of best-fitting ERGM for the empirical Avar data.

| Variable | Level | Fold | P-value | Type |
|---|---|---|---|---|
| Site | HNJ | 548.84 | $<10^{-4}$ | Diff |
| Site | KFJ | 857.23 | $<10^{-4}$ | Diff |
| Site | KUP | 691.48 | $<10^{-4}$ | Diff |
| Site | RK | 184.98 | $<10^{-4}$ | Diff |
| Period | Early | 5.16 | $<10^{-4}$ | Diff |
| Period | Middle/Late | 9.17 | $<10^{-4}$ | Match |
| Sex/Age | Adult F/Adult F | 0.256 | $<10^{-4}$ | Mix |
| Sex/Age | Adult M/Adult M | 6.62 | $<10^{-4}$ | Mix |
| Sex/Age | *Adult F/Subadult F* | 0.89 | 0.753 | Mix |
| Sex/Age | Adult M/Subadult F | 7.29 | $<10^{-4}$ | Mix |
| Sex/Age | Subadult F/Subadult F | 4.40 | 0.009 | Mix |
| Sex/Age | *Adult F/Subadult M* | 1.18 | 0.274 | Mix |
| Sex/Age | Adult M/Subadult M | 7.12 | $<10^{-4}$ | Mix |
| Sex/Age | Subadult F/Subadult M | 9.14 | $<10^{-4}$ | Mix |
| Sex/Age | Subadult M/Subadult M | 12.8 | $<10^{-4}$ | Mix |
| Orientation | Orientation Match | 2.86 | $<10^{-4}$ | Match |
| Belt/Harness | Both Carry | 2.56 | $<10^{-4}$ | Diff |
| Belt/Harness | One Carries | 0.64 | $<10^{-4}$ | Diff |

The base case for comparison for the Age/Sex variable was Adult F/Adult M. Coefficients that are not significant are in italics. match = Node Match, Diff = Differential Node Match and Mix = Node Mix.

after they reached some age of maturity, that they left behind both living brothers, and siblings who had died before this age was reached.

Variables from archaeological contexts were also included. Horse riding equipment (iron bits, belt and harnesses) yielded a significant node-match (differential) relationship. We find that, compared to two individuals both buried without horse riding equipment, pairs where only one individual was buried with the equipment were significantly less likely to be related (0.64-fold).

In contrast, if both individuals were buried with horse riding equipment, then they were significantly more likely to be related (2.56-fold). Since horse riding equipment has been suggested to be an burial item indicating high status in Avar society at these sites, it may be the case that the probability of relatedness decreasing only when one individual has the item may be evidence that unions rarely crossed barriers due to "social status". By contrast, iron buckles were not significant. This indicates that these items were either "randomly" given to individuals (with respect to genetic relatedness), or were correlated with other variables (such as period or site), and hence not statistically meaningful.

Finally, burial orientation was significant, and individuals with matching burial orientation had an approximately 2.86-fold increase in relatedness. This finding supports the observation of Gnecchi-Ruscone *et al.* that different funerary customs are associated with different sub-groups within the overall pedigree, an association between archaeological context and fine-scale genetic affinity that our method identified. However, we cannot dismiss that orientation is correlated with individuals buried closely in time, and may just represent subtle changes observed over fine-scale time periods.

## Discussion

ERGMs are a powerful method for performing regression on network connectedness, allowing researchers to combine multifaceted variables, from genetics, archaeology, anthropology and other sources, into one analysis. The simple output from these regression models allows researchers to identify variables of importance and to quantify their impact on the network, allowing for intuitive, real-world interpretations of the final model. We see that future work could integrate uncertainty in the IBD networks directly into the ERGMs framework, and we encourage researchers to

follow strict quality control when constructing networks of relatedness estimates (either via IBD or other estimators) to mitigate these potential effects.

We have shown that using the Bayesian Information criterion is the best method for model selection allowing for the comparison of competing models of varying complexity. By investigating the limits of BIC to detect variables of interest with varying effects, it can be seen that ERGMs can detect even very subtle variables, and report the resolution of the statistical power for varying network sizes. Future work towards implementing dyad-dependent ERGMs, through addressing issues of computational feasibility and model degeneracy might lead to improvements in model precision and statistical power (Karwa et al. 2022).

The utility of ERGMs in this setting was demonstrated by reanalyzing the Avar network reported by Gnecchi-Ruscone et al. We were able to recreate the key result by showing that genetic sex was a significant predictor for network connectedness. We were also able to show this while accounting for site of origin and time period, both which should affect connectedness, but which could have been correlated with sex due to sampling bias. Further, we were able to show that the age at death was also a significant predictor, further reinforcing the finding of a patrilineal kinship system, in which patrilocality and female exogamy were the norm. We also report a new finding where body orientation was a significant predictor of relatedness.

Overall, as relatedness networks become more common in archaeogenetic studies, we present ERGMs as a proven method for understanding their structure. As in all studies of the human past, it is critical that multiple sources of information are co-analyzed, so that the most robust interpretations of the data can be found. ERGMs inherently encourage this approach to interdisciplinary analysis in a statistically rigorous and familiar framework.

## Data availability

All sequence data have been deposited in the European Nucleotide Archive (ENA) with the accession number PRJEB72021, and all metadata can be found with the original publication (Gnecchi-Ruscone et al. 2024).

The code used to perform the simulation study can be found at https://github.com/jonotuke/ancient-ergms and the code used to perform all other analyses in R can be found at https://github.com/BenRohrlach/ERGMsForIBDAnalysesPaper.

Supplemental material available at GENETICS online.

## Acknowledgments

The authors wish to thank Prof. Johannes Krause for his support in this research, and Dr. Harald Ringbauer, Professor Nigel Bean, and Dr. Vincent Braunack-Mayer for enlightening and instructive conversations.

## Funding

ABR was supported by the Max Planck Society. This research was funded by the European Research Council under the European Union's Horizon 2020 research and innovation program under grant agreement numbers 101141408-ROMANCE and 856453-HistoGenes.

## Conflicts of interest

The authors have no conflict of interest to declare.

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

*Editor: N. Barton*