## [Peer Review File · Genetics]

Detecting and Quantifying Networks of Biological Kinship via Exponential Family Random Graph Models

Adam Rohrlach, Guido Gneccchi-Ruscione, Zuzana Hofmanová, Zsófia Rácz, Matthew Roughan, Wolfgang Haak, and Jonathan Tuke

NOTE: The reviews and decision letters are unedited and appear as submitted by the reviewers.

In extremely rare instances and as determined by a Senior Editor or the EIC, portions of a review may be redacted. If a review is signed, the reviewer has agreed to no longer remain anonymous.

The review history appears in chronological order.

Review Timeline:

Submission Date:	2025-09-08
Editorial Decision:	2025-11-18
Resubmission Received:	2025-12-08
Accepted:	2026-02-17

November 17, 2025

GENETICS-2025-308571

Detecting and Quantifying Networks of Biological Kinship via Exponential Family Random Graph Models

Dear Dr. Rohrlach:

Three experts in the field have reviewed your manuscript, and I have read it as well. While your manuscript is not currently acceptable for publication in GENETICS, we would reconsider a substantially revised manuscript. The reviewers have comments and concerns to be addressed in a revised manuscript. You can read their reviews at the end of this email.

The reviewers provide several suggestions to improve the paper. A major concern is the one raised by reviewer 1 (and elaborated on in my comments) regarding how valid the core modeling assumptions in EGRMs are for use with genetic data where relatedness is not independent due to background genetic/family structure that is unobserved (i.e. not captured by observed covariates). We look forward to receiving your revised manuscript. Please let the editorial office know approximately how long you expect to need for revisions.

Upon resubmission, please include:

1. A clean version of your manuscript;
2. A marked version of your manuscript in which you highlight significant revisions carried out in response to the major points raised by the editor/reviewers (track changes is acceptable if preferred);
3. A detailed response to the editor's/reviewers' feedback and to the concerns listed above. Please reference line numbers in this response to aid the editor and reviewers.

Your paper will likely be sent back out for review.

Additionally, please ensure that your resubmission is formatted for GENETICS
<https://academic.oup.com/genetics/pages/general-instructions>

Follow this link to submit the revised manuscript: Link Not Available

Sincerely,

John Novembre
Associate Editor
GENETICS

Approved by:
Nicholas Barton
Senior Editor
GENETICS

Reviewer #1 :

See attachment.

Reviewer #2 :

Disclaimer: I am an expert in archaeology and network science, and my review focuses on these fields and not others.

This paper presents original work and results that build on previous empirical work, with the main contribution being the application of EGRMs to networks derived from ancient DNA. The empirical work builds on a previously published network reconstruction and significantly enhances it by focusing on complex network dependencies. The authors demonstrated the ERGM approach's ability to replicate the original study's findings, as well as reporting a new finding concerning body orientation.

I believe the paper is clear, focused, concise, well-written and convincing. I did not identify major issues in my own areas of expertise, and I believe two factors demonstrate novelty: the first application of EGRMs to ancient DNA networks, and the new result concerning body orientation.

A few minor comments:

Section A describes a cutoff value c which raised some questions in me. In an applied case, ERGMs take an empirical network as input and results are sensitive to the precise edges that are present or absent. I don't believe this sensitivity or the cutoff value are discussed in the simulation study, or in the applied empirical study. Section E1 in the supplements describes what constitutes a relationship, and this decision (and therefore the precise empirical network structure) was taken from the previous empirical work. This is appropriate since this paper explicitly builds on, replicates and expands this previous work. However, the current paper on its own does not discuss or demonstrate the appropriateness of the edge definition and cutoff, or how these decisions impact the results of ERGMs. I raise this because the empirical networks derived from ancient DNA are not a given, they are the result of making a decision for a certain measure of what defines similarity and using a cut-off value. These decisions determine the resulting network structure, which is therefore sensitive to cutoff value definition. Perhaps an explicit statement could be added in the empirical case study about this, and/or the method section could discuss or demonstrate the sensitivity of ERGM results on edge cutoff value?

The discussion of the overfitting issue and the use of BIC are appropriate. However, the paper on its own did not clarify particularly clearly what models were selected and compared, and what theory or previous work supports these models being plausible (at some point reading the main text I thought incorrectly that BIC was used to navigate the entire model landscape rather than initial model selection being theoretically motivated). Perhaps this could be made more clear with a strategic sentence here or there.

Fig. 5: a lot of variables are shown in this figure. The node size differences in the orientation is not very clear or intuitive to understand for me, and I could not identify what the grey and black colours for period stood for (the edges are grey, so perhaps it concerns the edges, but I do not see any black edges).

The paper considers networks constructed from ancient DNA as well as archaeological data. ERGMs have been applied to archaeological data before, and previous work on the combination of ancient DNA and archaeology networks also exists. Perhaps the authors can consider citing some of these to more clearly position the contribution tightly on the application of ERGMs to ancient DNA networks rather than archaeology as a whole:

Amati, V. (2024). Random graph models. In Brughmans, Tom, B. J. Mills, J. L. Munson, & M. A. Peeples (Eds.), *The Oxford Handbook of Archaeological Network Research*. Oxford: Oxford University Press.

Amati, V., Mol, A., Shafie, T., Hofman, C., & Brandes, U. (2019). A Framework for Reconstructing Archaeological Networks Using Exponential Random Graph Models. *Journal of Archaeological Method and Theory*. <https://doi.org/10.1007/s10816-019-09423-z>

Brughmans, T., Keay, S., & Earl, G. (2014). Introducing exponential random graph models for visibility networks. *Journal of Archaeological Science*, 49, 442-454. <https://doi.org/10.1016/j.jas.2014.05.027>

Mazzucato, C., Coscia, M., Küçükakdağ Doğu, A., Haddow, S., Kılıç, M. S., Yüncü, E., & Somel, M. (2025). "A Network of Mutualities of Being": Socio-material Archaeological Networks and Biological Ties at Çatalhöyük. *Journal of Archaeological Method and Theory*, 32(1), 25. <https://doi.org/10.1007/s10816-024-09692-3>

Wang, L.-Y., & Marwick, B. (2021). A Bayesian networks approach to infer social changes from burials in northeastern Taiwan during the European colonization period. *Journal of Archaeological Science*, 134, 105471. <https://doi.org/10.1016/j.jas.2021.105471>

Reviewer #3 :

This paper presents a framework for applying ERGMs to kinship networks derived from ancient DNA combined with archaeological data. The authors use simulations to evaluate how well ERGMs can identify the covariates that influence the probability of genetic relatedness between individuals, comparing different model-selection strategies and quantifying statistical power. They apply the approach to an Avar-period dataset from the Carpathian Basin, recapitulating previous results and providing some additional insights. The study highlights the potential usefulness of ERGMs to detect subtle drivers of genealogical structure in ancient populations. Overall, the paper describes an elegant application of a network-based method to a relevant problem in paleogenomics, with all the elements required from a methods paper - theoretical description, simulation analysis, power analysis and a case-study demonstration. There are a few points where the paper requires improvements in my opinion, which I detail below.

It is a shame that line numbers are missing, making it difficult to review the paper. Equation numbering could also be helpful.

Major

One area where some analysis is missing is in regard to the cutoff, c . Because the method takes the weighed relatedness matrix and transforms it to an undirected unweighted network using the cutoff, there is a major question raised regarding the dependence of the results on the cutoff c . I understand the need to simplify the relatedness network, and the authors briefly

touch on this on page two left column, but there is no evaluation procedure that describes how the choice of c can affect the results. How robust are conclusions to changes in this cutoff value? This is not a new problem in the literature, with previous works on individual genetic similarity networks and IBD-sharing networks showing that playing with this cutoff, even when preserving edge weights above the cutoff, can substantially affect results of analysis (mostly population structure analysis). Moreover, informed choice of this cutoff can direct the analysis towards a specific research question (e.g. focusing only on very recent kinship vs. deep relatedness, or maybe even specific kinship classes). Addressing this question, in the context of previous literature, and adding appropriate analyses that demonstrate the extent to which this cutoff can be adjusted without changing conclusions, is important.

This leads me to the next point - there is a gap in how previous work is described and in contextualization of this work in light of the literature. There have been many works on analysis of genetic-similarity/relatedness/IBD-sharing networks; it is important to show connections between this work and previous works so that we can draw ideas and methods from previous efforts. The usefulness of ERGMs can probably be shown in other fields beyond aDNA analysis, and drawing these connections could help.

Another issue that deserves some discussion is that p_{ij} depends only on the covariates of i and j , so the paper only discusses dyad-independent ERGM. It should be clearly noted that the framework presented here adopts this (sensible) limitation, but it would be interesting to discuss when we should consider general ERGMs with relationship between multiple edges (transmissible cultural practices?).

Minor

Table 1 - The caption states that the base sex for comparison was adult F/Adult F, but the fold change for this pairing in the table is given as 0.256. Shouldn't it just be one, if this is the case you are comparing to? Is this a typo somewhere?

Bullet points in page 2 are unnecessary

Page 2A, left column - typo v_i, v_i should be v_i, v_j

Page 3B - "it is more naturally matched to the type of models being used here" - please explain why

Page 3B - I'm not sure that the description of AIC and BIC in these terms (e.g., the discussion of the difference of 2 in BIC/AIC guidelines) is really required. What is important to understand is what procedure you used and why.

Page 3D - why Louvain algorithm? This is a fast and efficient algorithm, but is also stochastic (did you look at multiple runs?) and does not have the best resolution. If my network is not very large, which would be the case in many cases in aDNA, why not used more powerful but deterministic and slower algorithms?

Page 3C - The description of the basic simulation setup here is insufficient. Please describe what exactly you simulated and how.

Fig. 4 - I like the power analysis here, it provides good intuition. However, I am confused why the units on the x-axis are the number of nodes, whereas in this dyad-independent setup the power should depend on the number of edges (or be proportional to the number of nodes²), as the authors point out themselves on page 3 C.

Page 4F - what was the c cutoff for the Avar network? How was it determined? Would conclusions change if I choose a different value?

Page 4F - typo "Section ??"

Supplement, B5 - Looks like an error here, if $\theta_6 = \theta_7 = 3$ than shouldn't the fold-increase be around 15, not 2? Maybe it should be $\theta_6 = \theta_7 = 0.75$?

Supplement C, second equation - what is θ_{0+} ? Looking at the math, this seems like another typo.

For discussion - How can we account for uncertainty in allele-sharing and relatedness calls? This is of particular importance in aDNA, and given that a strict related/unrelated cutoff is used which could lead to sequencing errors being amplified. This could lead to overconfidence in the model selection step. Maybe we can integrate over uncertainty outputs of the relatedness software? Or propagate uncertainty to the ERGM estimates?

Data Availability - where is the code for generating the analyses and figures in this paper deposited?

Associate Editor Comments:

As my own expression of reviewer 1's comment -- analysts in statistical genetics/phylogenetics have had to contend with background genetic structure for decades. While different in application area - this recent paper by Schraiber, Edge, and Pennell is a useful review of the issues in a unified framework:

Schraiber JG, Edge MD, Pennell M. Unifying approaches from statistical genetics and phylogenetics for mapping phenotypes in structured populations. *PLoS Biol.* 2024 Oct 9;22(10):e3002847. doi: 10.1371/journal.pbio.3002847. PMID: 39383205; PMCID: PMC11493298.

The concern for your manuscript is that the use of EGRNs in a straightforward way here will suffer in similar ways (albeit in new way given the edge-based data and model). While it's acknowledged here that the pairwise nature of the modeling is different from logistic regression assuming independence of observations - the model assumes an independence among pairs where in fact we should expect departures from the mean behavior will be highly correlated among genetic relatives. Ideally - you can examine these issues and attempt to address them, as the concern is that, as in other cases of unmodeled background

structure, they may cause false positives in the wrong settings.

To provide an example - suppose a family with one set of siblings of size 'n' is buried all with the same orientation - say $n=6$. Now let's double the 'n' to $n=12$ - should we take this as twice the evidence in favor of orientation matches leading to IBD matches? (Presumably not as in either case it's really just 1 family making a choice of an orientation and then a set of correlated observations). Or let's suppose we observed 3 families with $n=2$, where each pair of buried siblings was buried in the same orientation - would we take that as equal to the original? (Presumably not as the 3 families would represent perhaps 3 independent choices of orientation for the siblings rather than 1).

Summary

This manuscript attempts to apply Exponential Random Graph Models (ERGMs) to networks of genetic relatedness derived from ancient DNA, using a dataset from Avar-associated cemeteries as a case study.

There are scientific arenas in which network models are useful. This is not one of them. Reasons for this are given below under “Major Concerns.” Minor editorial comments which would improve the readability of the paper are given in “Minor Concerns,” organized by section.

Major Concerns

1. Inadequate justification and testing of the ERGM model

Nowhere do the authors address the potential problems with their approach introduced by non-independence. Instead, they simulate data from an independent-edges logit model. Since the only predictors used in their ERGM are node features, the model cannot handle dependence between edges. This is a fatal flaw of an IBD network, since many individuals are closely related (the pedigrees in Figure 2 of Gnecci-Ruscione et al., 2024 span 9 generations, with several lineages represented by 5 generations).

In the supplement they state, “We defined individuals as “related” if they shared at least two blocks of IBD of length 12cM, and at least one block of IBD of length 16cM.” Since the definition of an edge is the most important modeling decision to be made in network analysis, this definition should be in the main text, and they should justify the definition. The distribution of IBD blocks is determined by population structure and recombination (which is *not* a logit distribution). There are simulators available for generating IBD, and a network modeling approach cannot be justified without (a) simulating a process that generates IBD, (b) constructing networks from those IBD values, then (c) fitting their logit model to *those* networks.

In the empirical section, there should be some discussion of the IBD data used to construct the graph. What is the average IBD? How was IBD calculated? A network is simply a sparse covariance matrix, so if the IBD block length distribution is not sparse, there is no reason to build a network. If you can do “a network regression to model the log-odds of any two individuals being connected given the measured covariates” (pg 2), you can also do (for example) a multivariate regression on the fraction of IBD between individuals which – given the small size of the graphs under consideration – is probably more sensible.

Furthermore, in this dataset the reported coverage is 2.6x. This is quite low, so there is a *lot* of sequencing error. Thus IBD is likely underestimated. Since they intend to use their method for ancient DNA, the authors should have conducted simulations to determine the effect of underestimating IBD on their inferences.

In section A, they state “choice of cutoff might be determined through a statistical model of the data-measurement process, or by calibration to some ‘ground-truth’ data, for instance, obtained through in silico simulations. We then treat our network as a random sub-sample of the complete network, the sub-sample being determined via measurement noise, and the presence or absence of measured individuals” – however, the choice of cutoff is always arbitrary, since the proportion of shared IBD is a continuous value. Noise means you’re trying to recover some ground truth thing – here IBD *is* the ground-truth. When one builds a network, one is merely constructing a (biologically meaningless) coarsening of this ground-truth data.

2. No meaningful analysis of empirical data

The authors cite Gnecci-Ruscione et al. (2024), then continue with “the authors found that” and lists a string of observations. They should clarify where the observations of Gnecci-Ruscione et al. (2024) end and those of the current paper begin. Is the network topology the only aspect they are quoting from that paper? Or is this entire empirical section just a recapitulation of that work?

More importantly, the questions they address can all be answered by techniques other than ERGM which (unlike ERGM) have some basis in biological reality. For example, Gnecci-Ruscione et al. (2024) assign male genomes to Y haplogroups and use KIN and BREAD for to build pedigrees. This – unlike a logit model for network edges – is *real* evidence for female exogamy. Similarly, Gnecci-Ruscione et al. 2024 use ADMIXTOOLS and DATES to model admixture. These are well-tested tools underpinned by genetic models with justifiable assumptions. It is hard to see how the dubious methodology introduced in the present paper adds anything to this literature.

Minor Concerns

Introduction.

“we use simulated networks and Bayesian Information Criterion based model selection to show that we can reliably recover the correct, and hence best-fitting” this sentence is awkward. Model selection, by definition, recovers the best-fitting model. Conversely, in real statistical settings the best-fitting model is NEVER the correct model.

Section A.

“We can expand this approach” – moving from IBD to pedigrees is not expanding this approach, it is a completely different approach, as pedigrees and IBD values are very different types of data.

The terms nodematch, differential sitematch, and nodemix are introduced without being defined.

In this section they state “homophily (nodematch) is the primary factor determining connectedness, that is, a pair of nodes that share an attribute are more likely to be connected. An example of this may be that individuals that share the same burial site are more likely to share a genetic connection.” This formulation is causally backward. Homophily, in the network science literature, is a hypothesis to be tested. In an IBD network, I am buried in the same village and have similar age as my sister *because we share ancestry*.

“whether the attributes match” should be “how similar the attributes are” as most of the node attributes are real-valued.

Discussion of the results should use terminology consistent with the phenomenon being model, i.e. “that site A was more inbred than sites B or C” instead of “site A was more interconnected than sites B or C.”

Section B.

“more complicated (more highly parameterised models)” should be “more complicated (more highly parameterised) models”.

What does it mean that BIC “is more naturally matched to the type of models being used here”? if this is explained in reference 20, a few sentences here summarizing that argument should be inserted here.

BIC assumes true model belongs to set of candidate models, so this comparison only makes sense in the toy examples simulated here. In a more realistic simulation scheme, one would expect that AIC_c would outperform both BIC and AIC.

Section D.

If the goal is to do Louvain clustering on a network “weighted by the total amount of shared IBD,” really doesn’t make sense to first threshold the network by the criterion of “two blocks of IBD of length 12cM, and at least one block of IBD of length 16cM.”

Section E.

“the accuracy of model selection, and hence variable selection...”: we know at this point what model selection is, as this was explained in section B.

The authors use the terms “factors,” “covariates,” and “variables” willy-nilly throughout. While most readers will recognize these as synonyms, greater consistency would enhance readability.

“power is the expected 0.05”- what is this supposed to mean? In the same paragraph, they report power of “0.05,” “one,” and “80%”. Again, notation and units should be consistent.

Section F.

Fix the reference to “(see Section ??)”

“more inter-related” should be “have a higher inbreeding coefficient”

The authors note that “the age/sex of an individual was the only variable for which the more complex node-mix was selected” and use this to argue that their analysis supports patrilocality / female exogamy. Any measure of relatedness derived from IBD will be correlated with sex, unless the sex chromosomes are explicitly excluded from the IBD calculations. If this was done it is not mentioned in the paper.

November 17, 2025

GENETICS-2025-308571

Detecting and Quantifying Networks of Biological Kinship via Exponential Family Random Graph Models

Dear Dr. Rohrlach:

Three experts in the field have reviewed your manuscript, and I have read it as well. While your manuscript is not currently acceptable for publication in GENETICS, we would reconsider a substantially revised manuscript. The reviewers have comments and concerns to be addressed in a revised manuscript. You can read their reviews at the end of this email.

The reviewers provide several suggestions to improve the paper. A major concern is the one raised by reviewer 1 (and elaborated on in my comments) regarding how valid the core modeling assumptions in EGRMs are for use with genetic data where relatedness is not independent due to background genetic/family structure that is unobserved (i.e. not captured by observed covariates). We look forward to receiving your revised manuscript. Please let the editorial office know approximately how long you expect to need for revisions.

Upon resubmission, please include:

1. A clean version of your manuscript;
2. A marked version of your manuscript in which you highlight significant revisions carried out in response to the major points raised by the editor/reviewers (track changes is acceptable if preferred);
3. A detailed response to the editor's/reviewers' feedback and to the concerns listed above. Please reference line numbers in this response to aid the editor and reviewers.

Your paper will likely be sent back out for review.

Additionally, please ensure that your resubmission is formatted for GENETICS

<https://academic.oup.com/genetics/pages/general-instructions>

Follow this link to submit the revised manuscript:

<https://genetics.msubmit.net/cgi-bin/main.plex?el=A5NR1HYB6A5KKc7I3A9ftdSmLfH7chSPQnH2dvqjBwZ>

Sincerely,

John Novembre
Associate Editor
GENETICS

Approved by:
Nicholas Barton
Senior Editor
GENETICS

We thank the Editor and Reviewers for their constructive feedback and the opportunity to revise and resubmit our manuscript. We have carefully revised the paper to address all comments through additional analyses, clarifications, and improved presentation. We have also transferred the manuscript to use the Genetics template from Overleaf to best match the layout and referencing style. We believe that these revisions have significantly strengthened and improved the paper.

Reviewer #1 :

Summary

This manuscript attempts to apply Exponential Random Graph Models (ERGMs) to networks of genetic relatedness derived from ancient DNA, using a dataset from Avar-associated cemeteries as a case study.

There are scientific arenas in which network models are useful. This is not one of them. Reasons for this are given below under “Major Concerns.” Minor editorial comments which would improve the readability of the paper are given in “Minor Concerns,” organized by section.

We thank Reviewer 1 for taking their time to review our manuscript. Naturally, we disagree, and give our reasons below.

Major Concerns

1. Inadequate justification and testing of the ERGM model

Nowhere do the authors address the potential problems with their approach introduced by non-independence. Instead, they simulate data from an independent-edges logit model. Since the only predictors used in their ERGM are node features, the model cannot handle dependence between edges. This is a fatal flaw of an IBD network, since many individuals are closely related (the pedigrees in Figure 2 of Gnecci-Ruscione et al., 2024 span 9 generations, with several lineages represented by 5 generations).

We concede that through using only nodal attributes, we only consider dyad-independent ERGMs, which while not edge-dependent, are node-dependent and conditionally-independent, and hence have very different properties compared to Erdős–Rényi random graphs. To improve clarity on this limitation, we have added this sentence to address this in the main text: “It must be noted that this allows us to consider only dyad-independent ERGMs, and we make this choice for the reasons of computational efficiency, due to potentially large numbers of nodes in

the network, and model degeneracy due to the likely relative sparsity of the network.” (Lines 68-74).

Both Reviewers 2 and 3 are supportive of the use of our dyad-independent approach, and Reviewer 3 specifically notes that our decision to consider only dyad-independent models is a “sensible” limitation. This limitation is due to the fact that our adjacency matrix is relatively sparse (not everyone can be related to many, many people), and hence the MCMC approach to estimating coefficients is unstable in nearly all cases that we have tested, leading to model degeneracy. We believe that this sparsity likely renders biases (due to dyad-dependency) to be minor (and we now explore this). However, further to this “limitation” of sparsity, if the matrix were not sparse, then the run time of the MCMC step would be enormous for large networks with many variables, such as in IBD networks.

We have added a new simulation study to evaluate the effect of dyad-dependence. We estimated the dyad-dependence via an Approximate Bayesian Computation analysis of the empirical data (added to supplementary) to estimate this effect in a statistically rigorous manner, and have now simulated empirically-reasonable dyad-dependent networks. We then analysed these newly simulated networks via the approach presented in our study. We find that variable selection is still 100% accurate (the correct variables are still always recovered), but that the nature of homophily can be subtly misclassified. This only occurred in 2% of all simulations, and only when:

- Model 1 (site match) classified as Model 2 (differential site match),
- Model 2 (differential site match) classified as Model 1 (differential site match),

meaning that the only incorrect qualitative inferences made from model selection would be to assume that sites were more, or less, interconnected compared to one another than was true.

Reviewer 3 also adds that it would be interesting to discuss when we should consider general ERGMs with relationships between multiple edges, with which we agree. Hence we add “Future work towards implementing dyad-dependent ERGMs, through addressing issues of computational feasibility and model degeneracy might lead to improvements in model precision and statistical power.” to the conclusions. This is future work that we are excited about, but with the methods currently not readily available, is outside the scope of this study.

Our approach presents the first example of applying a parametric analysis approach to IBD networks, and is an important first step in the analysis of networks of IBD sharing. This allows for a feasible and direct analysis of the observed networks that archaeogeneticists are currently working with, and can only be improved as computational infeasibility and model degeneracy are accounted for in future work in the field of network analyses. We are excited to see steps forward in DERGMs and conditional edge ERGMs in this context.

In the supplement they state, “We defined individuals as “related” if they shared at least two blocks of IBD of length 12cM, and at least one block of IBD of length 16cM.” Since the definition

of an edge is the most important modeling decision to be made in network analysis, this definition should be in the main text, and they should justify the definition.

We agree that this is important, and we do mention it (i) when defining the cut off in the supplementary text and (ii) in the caption of Figure 5 in the main text, and we state (iii) that we are analysing the exact networks constructed by Gneccchi-Ruscione and colleagues who justify this cut off in their manuscript. We have added to the main text “where individuals share an edge if they share at least two blocks of IBD of length $\geq 12\text{cM}$ and at least one block of IBD of length $\geq 16\text{cM}$, indicating a degree of relatedness of approximately at least seventh-degree.” (lines 12-16).

The distribution of IBD blocks is determined by population structure and recombination (which is not a logit distribution). There are simulators available for generating IBD, and a network modeling approach cannot be justified without (a) simulating a process that generates IBD, (b) constructing networks from those IBD values, then (c) fitting their logit model to those networks.

We agree that our simulation methodology has some limitations. However, while there are simulators for generating IBD on a given pedigree, we know of no such method for simulating IBD that integrates demographic structure such as “one individual is more likely to select a mate with similar demographic attributes, such as burial goods or burial deposition properties”. However, the work of Ringbauer et al. 2023, Nature Genetics, has shown that the true discovery rate of IBD connections via ancIBD applied to imputed data performs well for sufficiently high-quality data, and with the appropriate quality control filters in place.

Hence, the simulation of the IBD to construct networks is beyond the scope of this study, as we are confident that a sufficiently representative, observed IBD-network has been recovered, and we directly simulate that network. Nevertheless, we included an additional simulation study showing that, if we randomly exclude observed edges with probability 0.05 (which is reasonable given the strict quality control thresholds), then our model selection results (via BIC) are unchanged. Hence, we show that our approach is resilient against a false negative rate as high as one in twenty edges.

We must also mention that we did not model IBD blocks via a logit distribution, but instead modeled their existence by using a stochastic Bernoulli distribution on the adjacency matrix. Nevertheless, to address the concern that the reviewer has with this approach, we have added an additional simulation study where simulations are produced via a network simulator that applies dyad-dependency, making edges dependent, for which we estimated the amount of dependency from the empirical data via Approximate Bayesian Computation (also mentioned in above in response to the dependency concern).

In the empirical section, there should be some discussion of the IBD data used to construct the graph. What is the average IBD? How was IBD calculated? A network is simply a sparse covariance matrix, so if the IBD block length distribution is not sparse, there is no reason to build a network. If you can do “a network regression to model the log-odds of any two

individuals being connected given the measured covariates” (pg 2), you can also do (for example) a multivariate regression on the fraction of IBD between individuals which – given the small size of the graphs under consideration – is probably more sensible.

Our study looked at the utility of ERGMs to analyse IBD networks, and not on the construction of such networks, nor the distribution of IBD on the network. It is outside of the scope of this work to explain how one should estimate IBD, as this is covered in quite some detail by Ringbauer et al. 2023 (among others), and we cite both this manuscript, as well as Gneccchi-Rusccone et al. who generated the data used in our study for their original manuscript.

However, we stress that we found a need in the literature, and in conversations with our colleagues in archaeogenetics, for a method to directly analyse the IBD networks that are being generated directly from empirical IBD data. Tools to directly analyse these networks together with contextual data are of significant interest to researchers, and that is the sole and main purpose of this study.

One could suggest insightful, alternative methods of analysis such as described by the Reviewer above, however if the network is sparse enough for such an analysis to pass the assumption of independence, then it is likely that dyad-dependency is also not so much of an issue. Instead, we advocate for researchers to be able to analyse their networks directly using an approach such as ERGMs. However, we also urge researchers to compare this to other, independent analyses (such as pedigrees, the inheritance patterns of uniparentally-inherited markers, spatially-embedded networks etc). It is for this reason that we included the statement “As in all studies of the human past, it is critical that multiple sources of information are co-analysed, so that the most robust interpretations of the data can be found.” in the conclusions.

Furthermore, in this dataset the reported coverage is 2.6x. This is quite low, so there is a lot of sequencing error. Thus IBD is likely underestimated. Since they intend to use their method for ancient DNA, the authors should have conducted simulations to determine the effect of underestimating IBD on their inferences.

AncIBD was used to estimate the pairwise IBD estimates, which is specifically designed for use with ancient DNA, and only attempts to call blocks with lengths of at least 8cM. Coverage of 2.6X is sufficiently high for ancient DNA SNP capture data, and well past the threshold for reliable IBD calling with ancIBD. Of note, quality control cut offs for being able to reliably apply ancIBD to ancient DNA have been well explored by Ringbauer et al. 2023, and we take their peer-reviewed work at face value. We apply the threshold of $\text{frac_gp} > 0.7$, indicating that (per individual) more than 70% of imputed sites met the quality control threshold of having posterior probability of > 0.99 . We have added this information (available in Gneccchi-Rusccone et al. 2023) to our supplementary section.

In section A, they state “choice of cutoff might be determined through a statistical model of the data-measurement process, or by calibration to some ‘ground-truth’ data, for instance, obtained

through in silico simulations. We then treat our network as a random sub-sample of the complete network, the sub-sample being determined via measurement noise, and the presence or absence of measured individuals” – however, the choice of cutoff is always arbitrary, since the proportion of shared IBD is a continuous value. Noise means you’re trying to recover some ground truth thing – here IBD is the ground-truth. When one builds a network, one is merely constructing a (biologically meaningless) coarsening of this ground-truth data.

This statement about the choice of cut off is a generalised statement for any network, and not specifically IBD networks. For IBD networks, the chosen cut-off is entirely subjective, but is a reasonable choice to explore networks of closely and distantly related individuals up to the ~6/7th degree of relatedness. Where exactly the cut off is set can be different from case study to case study, and to address this, we now state in the main text “in the case of IBD networks, the choice of c reflects the research questions of interest” (lines 36-37).

We present a method that allows the exploration of IBD networks, i.e. of biologically related individuals, in conjunction with meta data from archaeological and anthropological contexts, for which there is high demand in the field of archaeogenetics and archaeology to jointly analyse contextualised networks of biologically related individuals.

To suggest that networks of IBD are biologically meaningless is a strong personal opinion, which we will not address. Instead we would like to point the reviewer to a number of recent studies, such as Gerber et al. 2024 (Science Advances), Nordfors et al. 2025 (iScience), Szécsényi-Nagy et al. 2025 (Nature Communications), Gretzinger et al. 2025 (Nature), Wang et al. 2025 (Nature), among many others. These studies have produced meaningful results and interpretations, and showcase that the analysis of IBD networks is a useful and popular method for exploring aDNA data sets. Critically, our own study presents further method development for the analyses of IBD networks.

2. No meaningful analysis of empirical data

The authors cite Gnecci-Ruscione et al. (2024), then continue with “the authors found that” and lists a string of observations. They should clarify where the observations of Gnecci-Ruscione et al. (2024) end and those of the current paper begin. Is the network topology the only aspect they are quoting from that paper? Or is this entire empirical section just a recapitulation of that work?

The purpose of this study was never to produce a brand new analysis of the data. We instead aimed to show what a reanalysis of the data could yield. Specifically, we aimed to show that on empirical data, ERGMs can:

1. Recover the same findings as a peer-reviewed study (showing that the method works on empirical data analysed using more traditional methods).
2. Uncover “new” (or perhaps more nuanced) findings, showing that ERGMs can potentially uncover deeper signals in the data.
3. Be used as an exploratory tool that can help to decide on demographic or any other contextual variables that can be further investigated.

We feel that the IBD network of Avar individuals from Gnecci-Ruscione et al. perfectly fit these requirements.

More importantly, the questions they address can all be answered by techniques other than ERGM which (unlike ERGM) have some basis in biological reality. For example, Gnecci-Ruscione et al. (2024) assign male genomes to Y haplogroups and use KIN and BREAD for to build pedigrees. This – unlike a logit model for network edges – is real evidence for female exogamy. Similarly, Gnecci-Ruscione et al. 2024 use ADMIXTOOLS and DATES to model admixture. These are well-tested tools underpinned by genetic models with justifiable assumptions. It is hard to see how the dubious methodology introduced in the present paper adds anything to this literature.

We find it surprising that using additional methods to confirm core analyses and findings is not considered a positive aspect of our method. As statisticians, we encourage researchers to always use multiple, hopefully independent methods, with different strengths and weaknesses, to see if and when they agree or disagree. We cannot see how adding another confirmatory analysis is a weakness.

Further, we would like to point out that many studies that use network analyses do not have the luxury of containing so many individuals that produce fully articulated pedigrees. In cases where the degrees of relatedness separating sub-pedigrees are greater than two or three (depending on coverage/kin estimation), no unique pedigree even exists, and only sub-pedigrees can be found. However, using a network representation of the embedded pedigree with our method still allows for an analysis to be performed.

It is important to highlight that the work of our colleagues (three of whom are authors on this study) in analysing the relatedness/pedigrees on the original study took months of impressive, but painstaking work. By contrast, we were able to produce our pedigree-free analysis in less than an hour and produce similar findings. Moreover, the IBD network contains more distantly related individuals than can be shown/reconstructed in the/as pedigree, i.e 4th degree and higher relatives that cannot be estimated easily with KIN or similar methods. Critically, ERGMs allow us to analysed many variables at once and to co-explore the numerous variables in the data. Hence, simple exploratory ERGM analyses may be particularly useful if the variable of interest (e.g., burial item or burial position) was not obvious or expected from the context itself. For example, genetic sex might be an obvious variable to test, but a simple/quick ERGMs analysis can rapidly indicate which other variables may be of importance (as we did with burial position), either primarily or as lurking variables that must be controlled for, in a statistically rigorous way.

Adding to this, our method produces a quantified estimate of the effect of certain variables. For example, while Gnecci-Ruscione et al. show that genetic sex is a driver of network connectedness, we show that two adult female individuals are approximately four times less likely to be related than an adult male and an adult female.

Nevertheless, we must point out that the authors of the original manuscript use their own type of network analysis, although it is different from the one we propose here (on the same network that we analyse). The original paper uses the distribution of a network statistic, the degree of centrality, for (a) the whole network, (b) the network filtered to have only males, and (c) the network filtered to have only female individuals, to show that female exogamy explains the deeper connectedness of individuals from these sites. This was then viewed together with the Y-chromosomal diversity, and the pedigree reconstructed with BREADR and KIN, to produce this finding of female exogamy. The new approach presented here could have filled the need for this bespoke analysis.

In summary, in reanalysing the work of Gneccchi-Rusccone *et al.*, with authors from the original study as co-authors in this study, we make two important points about our method.

1. We were able to reproduce their results using our approach, while co-analysing many variables at once. This approach has the benefit that we are accounting for lurking variables that may have biased the results. For example, by taking into account genetic sex and site at the same time (among other concurrent variables) we can state that the pattern that was observed for female exogamy was not due to a genetic sex imbalance at the sites. This statement is true for other potential lurking variables that we included, such as the time period.
2. We found new results, while still supporting the original findings. The authors of the original study did not include the age-at-death of the individuals, meaning that our statement about female exogamy took into account that this only applies to *adult* female individuals, which one would expect if women were leaving their place of birth to join their partner only *after* reaching (potentially sexual) maturity. Furthermore, we find that horse riding equipment was a significant predictor of connectedness, which archaeologists had hypothesised may be an indicator of class stratification (for which there is no genetic marker). This finding was first tested in a later study of the Avar (Wang et al. 2025, Nature), and was then confirmed in this study due to these new analyses. We also identify the period and the burial position as significant variables, which were not included in the original network analysis.

Minor Concerns

Introduction.

“we use simulated networks and Bayesian Information Criterion based model selection to show that we can reliably recover the correct, and hence best-fitting” this sentence is awkward. Model selection, by definition, recovers the best-fitting model. Conversely, in real statistical settings the best-fitting model is NEVER the correct model.

We agree that this language was awkward, and in the wrong order. We have changed this to say “we use simulated networks and Bayesian Information Criterion based model selection to show that we can reliably recover the correct simulated model, hence uncovering known variables of interest.” (lines 10-13).

Section A.

“We can expand this approach” – moving from IBD to pedigrees is not expanding this approach, it is a completely different approach, as pedigrees and IBD values are very different types of data.

We did not mean to infer that pedigrees and IBD values are the same type of data, but instead meant to say that one could naturally move from “IBD networks to pedigrees”. We are merely stating here that if one knew the pedigree, and hence the direction of relatedness (i.e., who is the ancestor of who), one could use a directed network without having to fundamentally change the method. We have removed this sentence to avoid any confusion.

The terms *nodematch*, *differential sitematch*, and *nodemix* are introduced without being defined.

There was a typo here, and “*differential sitematch*” should have been “*differential nodematch*”. In lines 94-104 we introduce the ERGMs terms for simple homophily, differential homophily and attribute mixing (*nodematch*, *differential nodematch* and *nodemix*, respectively), with the terms given in brackets after the technical definition. For example, we say “It might be that simple homophily (*nodematch*) is the primary driver determining connectedness” to introduce the term “*nodematch*” via the true definition of “simple homophily”.

In this section they state “homophily (*nodematch*) is the primary factor determining connectedness, that is, a pair of nodes that share an attribute are more likely to be connected. An example of this may be that individuals that share the same burial site are more likely to share a genetic connection.” This formulation is causally backward. Homophily, in the network science literature, is a hypothesis to be tested. In an IBD network, I am buried in the same village and have similar age as my sister because we share ancestry.

In no case can we test causality in an observed experiment, such as here. We also do not believe that we make a statement about causality, but simply correlation, i.e., that individuals found buried in the same place are more often related than individuals found buried in different, more distant places. However, to avoid confusion we have changed this to “An example of this may be that individuals who share the same burial site are more likely to have been genetically related”, using the past tense to infer the more likely causal relationship (lines 88-90).

“whether the attributes match” should be “how similar the attributes are” as most of the node attributes are real-valued.

We disagree here, as in this study the node attributes are qualitative/categorical variables with restricted possible values/levels. For our empirical data for example, the genetic sex can *only* be male or female (as no chromosomal aneuploidies were found), and the sites can only be one of the four excavated sites. Hence, these variables match or they do not. The reviewer is correct that there can be real-valued variables also, that may not be exact at the measurement level.

For example, the direction of burial might not be identical, but once recorded N-N-W is not considered any closer to N-W than (say) S-W. So “match” is what is required for the method.

Discussion of the results should use terminology consistent with the phenomenon being model, i.e. “that site A was more inbred than sites B or C” instead of “site A was more interconnected than sites B or C.”

We are not referencing the inbreeding levels of any site. We simply mean that some sites may have more individuals that are more closely related than other sites (although inbreeding may be one potential cause for this). For example, if site A had a single pedigree, but site B is made up of many smaller, unrelated pedigrees, then site A may appear more connected within the site/interconnected (i.e. higher mean degree of centrality) than site B.

We have made this clearer by changing the example to read “For example, imagine that site A was made up of one pedigree, but sites B and C are made up of several, unrelated pedigrees. It may still be the case that individuals buried at the same site are more likely to share a genetic connection, but that since sampling site A will always produce pairs from the same pedigree, that site A was more interconnected than sites B or C. Hence, two individuals buried at site A are more likely to be connected than two individuals buried at site B (or C).” (lines 95-103).

Section B.

“more complicated (more highly parameterised models)” should be “more complicated (more highly parameterised) models”.

Thank you. This was a typo and has been corrected.

What does it mean that BIC “is more naturally matched to the type of models being used here”? if this is explained in reference 20, a few sentences here summarizing that argument should be inserted here. BIC assumes true model belongs to set of candidate models, so this comparison only makes sense in the toy examples simulated here. In a more realistic simulation scheme, one would expect that AICc would outperform both BIC and AIC.

AICc is specifically designed to correct the AIC for small sample sizes. AICc does this by introducing a penalty such that $AICc = AIC + (2k^2 + 2k)/(n - k - 1)$ where n is the sample size, and k is the number of parameters in the model. As n (the number of observed pairs) grows large, which happens quickly as it grows factorially with the number of nodes N , this penalty term quickly tends to zero. To show this, we added the AICc to the simulation study and found that AICc and AIC are perfectly correlated ($r=1$, $p \approx 0$) with a regression equation of the form $AICc = AIC$ (intercept 0 and slope 1, $p \approx 0$ for both). Hence, it is clear that AICc performs identically to AIC, and so we do not report it. We now state in the main text that “We also calculated the corrected AIC (AICc)(Sugiura 1978), but found that due to large number of observed pairs of nodes relative to the number of parameter values, that it performed identically to the AIC”, and have added a section to the supplementary text reporting this.

Conversely to AIC/AICc, BIC performs best in situations where the sample size is very large. As argued above, this is the case for ERGMs, where the sample size, defined by the number of node pairs, grows factorially as the number of nodes increases.

Finally, it is true that BIC assumes that the true model is one of the candidate models, but this is true of any Bayesian model selection method. We do not subscribe to the idea that only if the *exact* true model is in the candidate set, should we use BIC, as long as a sufficiently reasonable model is present. We do not think that it is the place of this paper to bring into question the approach of BIC. However, we do show that BIC performs best under simulation, and thus should be preferred.

Section D.

If the goal is to do Louvain clustering on a network “weighted by the total amount of shared IBD,” really doesn’t make sense to first threshold the network by the criterion of “two blocks of IBD of length 12cM, and at least one block of IBD of length 16cM.”

Measures of IBD on ancient samples are not free from statistical noise, and filtering edges via this threshold was of importance to avoid noise in the clustering process by removing edge weights that should likely be zero.

Section E.

“the accuracy of model selection, and hence variable selection...”: we know at this point what model selection is, as this was explained in section B.

We have removed “and hence model selection”.

The authors use the terms “factors,” “covariates,” and “variables” willy-nilly throughout. While most readers will recognize these as synonyms, greater consistency would enhance readability.

We have changed these cases to be the more general term “variable” in each case.

“power is the expected 0.05” - what is this supposed to mean?

When the fold-change is one, i.e. there is no change, we would expect the statistical power to be the same as the significance level since any significant result must be due to chance, and not true signal in the data. We have changed this to read “We find that when $\theta_1 = 0$, that is when there is no fold-change, power is the expected 5%, matching the significance level of $\alpha = 0.05$ ” (lines 74-76).

In the same paragraph, they report power of “0.05,” “one,” and “80%”. Again, notation and units should be consistent.

We thank the reviewer for pointing this out, and have changed all power statements to be in %.

Section F.

Fix the reference to “(see Section ??)”

Many thanks. We have fixed this error.

“more inter-related” should be “have a higher inbreeding coefficient”

Again, we mean that the average degree of centrality is higher, and make no statements about inbreeding coefficients in this study.

The authors note that “the age/sex of an individual was the only variable for which the more complex node-mix was selected” and use this to argue that their analysis supports patrilocality / female exogamy. Any measure of relatedness derived from IBD will be correlated with sex, unless the sex chromosomes are explicitly excluded from the IBD calculations. If this was done it is not mentioned in the paper.

We agree that this is true, and would never calculate IBD on the sex chromosomes for general population analyses. To make this clear we have added “IBD blocks (for only the autosomal chromosomes) were called using anclBD” to the Supplementary notes on data production.

Reviewer #2 :

Disclaimer: I am an expert in archaeology and network science, and my review focuses on these fields and not others.

This paper presents original work and results that build on previous empirical work, with the main contribution being the application of ERGMs to networks derived from ancient DNA. The empirical work builds on a previously published network reconstruction and significantly enhances it by focusing on complex network dependencies. The authors demonstrated the ERGM approach’s ability to replicate the original study’s findings, as well as reporting a new finding concerning body orientation.

I believe the paper is clear, focused, concise, well-written and convincing. I did not identify major issues in my own areas of expertise, and I believe two factors demonstrate novelty: the first application of ERGMs to ancient DNA networks, and the new result concerning body orientation.

We thank the reviewer for their constructive evaluation of our study.

A few minor comments:

Section A describes a cutoff value c which raised some questions in me. In an applied case, ERGMs take an empirical network as input and results are sensitive to the precise edges that

are present or absent. I don't believe this sensitivity or the cutoff value are discussed in the simulation study, or in the applied empirical study. Section E1 in the supplements describes what constitutes a relationship, and this decision (and therefore the precise empirical network structure) was taken from the previous empirical work. This is appropriate since this paper explicitly builds on, replicates and expands this previous work. However, the current paper on its own does not discuss or demonstrate the appropriateness of the edge definition and cutoff, or how these decisions impact the results of ERGMs. I raise this because the empirical networks derived from ancient DNA are not a given, they are the result of making a decision for a certain measure of what defines similarity and using a cut-off value. These decisions determine the resulting network structure, which is therefore sensitive to cutoff value definition. Perhaps an explicit statement could be added in the empirical case study about this, and/or the method section could discuss or demonstrate the sensitivity of ERGM results on edge cutoff value?

We thank the reviewer for their insights here. We first want to state that the choice of a cut off must be specific to the study. For example, if pedigree level patterns are of interest, then the network must be restricted to closer degrees of relatedness. If, instead, patterns of relatedness in a larger mating network were of interest, then c might be relaxed to uncover inter-site, cross-regional relationships. One could relax this even further if IBD sharing across large geographic distances and cultural backgrounds was of interest.

We trust that researchers will effectively select a sensible and informed value of c for their study. However, we have also added a sensitivity analysis to the empirical example presented in this study to indicate the effect of varying c (via the total sum of IBD blocks $\geq 8cM$). We show that feature selection appears unchanged, but that the nature of the homophily can subtly change, and that the coefficients may vary (for example, and quite predictably, a matching site was much more important as the degree of relatedness was made closer).

The discussion of the overfitting issue and the use of BIC are appropriate. However, the paper on its own did not clarify particularly clearly what models were selected and compared, and what theory or previous work supports these models being plausible (at some point reading the main text I thought incorrectly that BIC was used to navigate the entire model landscape rather than initial model selection being theoretically motivated). Perhaps this could be made more clear with a strategic sentence here or there.

We agree that it was not fully clear why we chose BIC. When introducing model selection measures, we have now added the sentence "Specifically, BIC is preferred when the sample size is much greater than the number of parameters in a model, which is the case for even relatively small networks where the number of potential edges grows factorially with the number nodes.", explaining the use of BIC. We have also added the corrected AIC (AICc) to the simulation study to show that this correction is not as empirically effective as BIC, and added this to the main text "We also calculated the corrected AIC (AICc) (Sugiura 1978), but found that, due to large number of observed pairs of nodes relative to the number of parameter values, it performed identically to the AIC". We have added a note to the supplementary titled "Comparing values of AIC to AICc in our simulations".

Fig. 5: a lot of variables are shown in this figure. The node size differences in the orientation is not very clear or intuitive to understand for me, and I could not identify what the grey and black colours for period stood for (the edges are grey, so perhaps it concerns the edges, but I do not see any black edges).

This plot was indeed quite busy. We have removed the additional details, and now present only the site and age/sex per node.

The paper considers networks constructed from ancient DNA as well as archaeological data. ERGMs have been applied to archaeological data before, and previous work on the combination of ancient DNA and archaeology networks also exists. Perhaps the authors can consider citing some of these to more clearly position the contribution tightly on the application of ERGMs to ancient DNA networks rather than archaeology as a whole:

Amati, V. (2024). Random graph models. In Brughmans, Tom, B. J. Mills, J. L. Munson, & M. A. Peeples (Eds.), *The Oxford Handbook of Archaeological Network Research*. Oxford: Oxford University Press.

Amati, V., Mol, A., Shafie, T., Hofman, C., & Brandes, U. (2019). A Framework for Reconstructing Archaeological Networks Using Exponential Random Graph Models. *Journal of Archaeological Method and Theory*. <https://doi.org/10.1007/s10816-019-09423-z>

Brughmans, T., Keay, S., & Earl, G. (2014). Introducing exponential random graph models for visibility networks. *Journal of Archaeological Science*, 49, 442-454. <https://doi.org/10.1016/j.jas.2014.05.027>

Mazzucato, C., Coscia, M., Küçükakdağ Doğu, A., Haddow, S., Kılıç, M. S., Yüncü, E., & Somel, M. (2025). "A Network of Mutualities of Being": Socio-material Archaeological Networks and Biological Ties at Çatalhöyük. *Journal of Archaeological Method and Theory*, 32(1), 25. <https://doi.org/10.1007/s10816-024-09692-3>

Wang, L.-Y., & Marwick, B. (2021). A Bayesian networks approach to infer social changes from burials in northeastern Taiwan during the European colonization period. *Journal of Archaeological Science*, 134, 105471. <https://doi.org/10.1016/j.jas.2021.105471>

We thank the reviewer for their important comment here. We did not mean to ignore the contribution of these studies to connecting network analyses to archaeology, and have added this sentence to the introduction to correct this "Network analyses, and specifically ERGMs, have also been used to analyse networks defined by archaeological data, making them a natural and familiar tool for analysing ancient DNA(19-23)".

Reviewer #3 :

This paper presents a framework for applying ERGMs to kinship networks derived from ancient DNA combined with archaeological data. The authors use simulations to evaluate how well ERGMs can identify the covariates that influence the probability of genetic relatedness between

individuals, comparing different model-selection strategies and quantifying statistical power. They apply the approach to an Avar-period dataset from the Carpathian Basin, recapitulating previous results and providing some additional insights. The study highlights the potential usefulness of ERGMs to detect subtle drivers of genealogical structure in ancient populations. Overall, the paper describes an elegant application of a network-based method to a relevant problem in paleogenomics, with all the elements required from a methods paper - theoretical description, simulation analysis, power analysis and a case-study demonstration. There are a few points where the paper requires improvements in my opinion, which I detail below.

We thank the reviewer for their positive description of our manuscript, and for their point-by-point evaluation of the requirements of a methodological study. We put a lot of emphasis on making sure that the statistical rigour of the method was addressed, and we thank the reviewers' suggestions to further strengthen this aspect.

It is a shame that line numbers are missing, making it difficult to review the paper. Equation numbering could also be helpful.

The lack of numbering was an unfortunate oversight. We were under the impression that equations should only be numbered if referenced in the text, but we can see how this is helpful when reviewing or reading a paper, and we have now added both.

Major

One area where some analysis is missing is in regard to the cutoff, c . Because the method takes the weighed relatedness matrix and transforms it to an undirected unweighted network using the cutoff, there is a major question raised regarding the dependance of the results on the cutoff c . I understand the need to simplify the relatedness network, and the authors briefly touch on this on page two left column, but there is no evaluation procedure that describes how the choice of c can affect the results. How robust are conclusions to changes in this cutoff value? This is not a new problem in the literature, with previous works on individual genetic similarity networks and IBD-sharing networks showing that playing with this cutoff, even when preserving edge weights above the cutoff, can substantially affect results of analysis (mostly population structure analysis). Moreover, informed choice of this cutoff can direct the analysis towards a specific research question (e.g. focusing only on very recent kinship vs. deep relatedness, or maybe even specific kinship classes). Addressing this question, in the context of previous literature, and adding appropriate analyses that demonstrate the extent to which this cutoff can be adjusted without changing conclusions, is important.

We thank the reviewer for their comments here. Each reviewer had the same concerns, and we repeat our response here (see also responses to Reviewers 1 and 2 above).

We believe that the choice of a cut off must be specific to the study. For example, if pedigree level patterns are of interest, then the network must be restricted to closer degrees of relatedness. If, instead, patterns of relatedness in a larger mating network were of interest, then

c might be relaxed to uncover inter-site relationships. One could relax this even further if continental, cross-cultural IBD sharing was of interest.

We trust that researchers will effectively select a sensible and informed value of c for their study. However, we have also added a sensitivity analysis to the empirical example presented in this study to indicate the effect of varying c (via the total sum of IBD blocks $\geq 8cM$) We show that feature selection appears unchanged, but that the nature of the homophily can subtly change, and that the coefficients may vary (for example, and quite predictably, a matching site was much more important as the degree of relatedness was made closer).

This leads me to the next point - there is a gap in how previous work is described and in contextualization of this work in light of the literature. There have been many works on analysis of genetic-similarity/relatedness/IBD-sharing networks; it is important to show connections between this work and previous works so that we can draw ideas and methods from previous efforts. The usefulness of ERGMs can probably be shown in other fields beyond aDNA analysis, and drawing these connections could help.

We thank the reviewer for this comment. We completely agree, and in addition to discussing the methods that pre-date IBD for relatedness, and the first studies that investigated what aDNA could say about demography, we now also modify a sentence about (i) the first studies to look at ancient DNA networks (“Archaeogenetic studies of relatedness networks have mostly focused on single network statistics calculated without taking into account potential network-wide dependency structures, and answer only limited hypotheses about single parameters, without quantifying these effects or considering potentially interacting variables”, lines 25-30) and add a new sentence about studies that have considered aDNA network clusters (“Other studies have applied clustering methods to explore embedded clusters of individuals to explore drivers of this clustering”, lines 31-34). Finally, as suggested by Reviewer 2, we have also added a sentence recognising the use of ERGMs in analysing networks constructed on archaeological data (“Network analyses, and specifically ERGMs, have also been used to analyse networks defined by archaeological data, making them a natural and familiar tool for analysing ancient DNA”, lines 43-46).

Another issue that deserves some discussion is that p_{ij} depends only on the covariates of i and j , so the paper only discusses dyad-independent ERGM. It should be clearly noted that the framework presented here adopts this (sensible) limitation, but it would be interesting to discuss when we should consider general ERGMs with relationship between multiple edges (transmissible cultural practices?).

The reviewer raises an important point, and for clarity we now state unequivocally that “It must be noted that this allows us to consider only dyad-independent ERGMs, and we make this choice for the reasons of computational efficiency, due to potentially large numbers of nodes in the network, and model degeneracy due to the likely relative sparsity of the network”. We feel that these concepts are beyond the scope of this paper, but will be an active area of research

once ERGMs are a familiar tool for the genetic research community, which we are excited about exploring in the near future.

We still felt that the effect of this was important to multiple reviewers, and so we have added a new simulation study to evaluate this important point, and we repeat here our response to Reviewer 1's similar criticism: We have added a new simulation study to evaluate the effect of dyad-dependence. We estimated the dyad-dependence via an Approximate Bayesian Computation analysis of the empirical data (added to supplementary) to statistically-rigorously estimate this effect, and have now simulated empirically-reasonable dyad-dependent networks. We then analysed these newly simulated networks via the approach presented in our study. We find that the correct variables are still always recovered, and that when models are misclassified, it was only involving site, and it was only in cases of nodematch having "differential" match levels or not (see supplementary section D).

Minor

Table 1 - The caption states that the base sex for comparison was adult F/Adult F, but the fold change for this pairing in the table is given as 0.256. Shouldn't it just be one, if this is the case you are comparing to? Is this a typo somewhere?

Yes, this was a typo. We thank the reviewer for spotting this, and we changed the caption to read "base sex for comparison was Adult F/Adult M, but the...".

Bullet points in page 2 are unnecessary

We agree and have removed the bullet points.

Page 2A, left column - typo v_i, v_i should be v_i, v_j

We have changed this to be correct.

Page 3B - "it is more naturally matched to the type of models being used here" - please explain why

We agree, and have added an additional sentence that reads "Specifically, BIC is preferred when the sample size is much greater than the number of parameters in a model, which is the case for even relatively small networks where the number of potential edges grows factorially with the number nodes". We also now explore the performance of AICc, which is functionally identical to AIC due to the large sample size (Supplementary D.1).

Page 3B - I'm not sure that the description of AIC and BIC in these terms (e.g., the discussion of the difference of 2 in BIC/AIC guidelines) is really required. What is important to understand is what procedure you used and why.

We believe that since information criteria themselves are estimates, that a concept of “significantly better” exists. Hence, we have added this requirement (as defined by Drton & Plummer) so that it is clear that we use this rule explicitly, and not when it suits us and is convenient. However, we have now moved this statement to the Supplementary description of model selection.

Page 3D - why Louvain algorithm? This is a fast and efficient algorithm, but is also stochastic (did you look at multiple runs?) and does not have the best resolution. If my network is not very large, which would be the case in many cases in aDNA, why not used more powerful but deterministic and slower algorithms?

We chose the Louvain algorithm as it is the default method currently used in the literature, and the clustering was just used to highlight the difference in inter-relatedness at the sites. This study does not address the best approaches to clustering on IBD networks, but we are certainly interested in exploring this in the near future.

Page 3C - The description of the basic simulation setup here is insufficient. Please describe what exactly you simulated and how.

We completely agree with this oversight, and have added a far more detailed section in the supplementary information on how the simulations were performed.

Fig. 4 - I like the power analysis here, it provides good intuition. However, I am confused why the units on the x-axis are the number of nodes, whereas in this dyad-independent setup the power should depend on the number of edges (or be proportional to the number of nodes²), as the authors point out themselves on page 3 C.

Thanks. We thought about this point carefully. Since the simulations are stochastic in nature, the number of edges was not constant, but was instead similar, between simulations. To estimate the power, we needed to repeat many simulations, and the only constant between these simulations was the number of nodes (even though the number of edges was consistently similar). We have changed this figure to have “the expected number of edges” on the x axis, and have included a supplementary section outlining the derivation of this expected value calculation.

Page 4F - what was the c cutoff for the Avar network? How was it determined? Would conclusions change if I choose a different value?

The cut off for the Avar network was 2 blocks of at least 12cM and one block of at least 16cM shared per pair. Since multiple reviewers asked this question, we realised that placing this in the text was important and have now added “where individuals share an edge if they share at least two blocks of IBD of length $\geq 12\text{cM}$ and at least one block of IBD of length $\geq 16\text{cM}$, indicating a degree of relatedness of approximately at least seventh-degree” to the empirical data description.

This value was chosen to match the value chosen by the authors of the original study, but we have now added a sensitivity analysis to the supplementary text to show the effect of changes in the relative cut off on the findings of the study.

Page 4F - typo "Section ??"

Thanks. This has been fixed.

Supplement, B5 - Looks like an error here, if $\theta_6 = \theta_7 = 3$ than shouldn't the fold-increase be around 15, not 2? Maybe it should be $\theta_6 = \theta_7 = 0.75$?

Yes. That was a typo and should have read that theta values were 0.75, not 3. This has been fixed..

Supplement C, second equation - what is θ_{0+} ? Looking at the math, this seems like another typo.

This was definitely a typo. Thanks for catching this.

For discussion - How can we account for uncertainty in allele-sharing and relatedness calls? This is of particular importance in aDNA, and given that a strict related/unrelated cutoff is used which could lead to sequencing errors being amplified. This could lead to overconfidence in the model selection step. Maybe we can integrate over uncertainty outputs of the relatedness software? Or propagate uncertainty to the ERGM estimates?

We do not look at statistical uncertainty in this first exploration of the use of ERGMs in IBD networks, and instead assume that the estimates are highly reliable. This is certainly an interesting area to explore and we add "We see that future work could integrate uncertainty in the IBD networks directly into the ERGMs framework, and we encourage researchers to follow strict quality control when constructing networks of relatedness estimates (either via IBD or other estimators) to mitigate these potential effects." to the conclusions to hopefully stimulate this discussion.

However, in this study, like in many published aDNA studies, we can best account for uncertainty by applying strict cut offs for quality control. We impute only samples with >500K SNPs on target for the 1240k array (indicating high coverage), apply strict mapping and sequencing quality thresholds, and then further filter individuals based on IBD quality control cut offs (which we do here, and have added a statement in the supplementary section reading "and who yielded a proportion of genotype posterior probabilities above the threshold of 0.99 for more than 70% of the imputed sites ($\text{frac_gp} > 0.7$)"). Following this, we apply the suggestions of the authors of ancIBD, the IBD estimation method used here that was designed specifically for ancient DNA, and do not use the lower bound of IBD (blocks of "just" 8cM) as evidence for IBD.

Data Availability - where is the code for generating the analyses and figures in this paper deposited?

Both Dr Tuke (simulation study) and Dr Rohrlach (remaining analyses) have created GitHub repositories for our code to be made public when the study is made public. We have added this line to the data availability statement “The code used to perform the simulation study can be found at <https://github.com/jonotuke/ancient-ergms> and the code used to perform all other analyses in R can be found at <https://github.com/BenRohrlach/ERGMsForIBDAnalysesPaper>”.

Associate Editor Comments:

As my own expression of reviewer 1's comment -- analysts in statistical genetics/phylogenetics have had to contend with background genetic structure for decades. While different in application area - this recent paper by Schraiber, Edge, and Pennell is a useful review of the issues in a unified framework:

Schraiber JG, Edge MD, Pennell M. Unifying approaches from statistical genetics and phylogenetics for mapping phenotypes in structured populations. PLoS Biol. 2024 Oct 9;22(10):e3002847. doi: 10.1371/journal.pbio.3002847. PMID: 39383205; PMCID: PMC11493298.

The concern for your manuscript is that the use of EGRNs in a straightforward way here will suffer in similar ways (albeit in new way given the edge-based data and model). While it's acknowledged here that the pairwise nature of the modeling is different from logistic regression assuming independence of observations - the model assumes an independence among pairs where in fact we should expect departures from the mean behavior will be highly correlated among genetic relatives. Ideally - you can examine these issues and attempt to address them, as the concern is that, as in other cases of unmodeled background structure, they may cause false positives in the wrong settings.

This is an important point, and one we now address more fully. We stress that we do not use a fully random graph, but instead assume node-dependence. The choice not to use dyad-dependent models was due to the fact that the networks we observe in empirical data are (relatively) too sparse for the MCMC estimation method for the dyad-dependent approach to converge (referred to as model degeneracy), but that if they were not so sparse, runtime for relatively small networks is far too long, and for large networks, is unable to be run on most modern hardware. Addressing these limitations is an active area of research, and we look forward to incorporating these leaps forward in methodology when they become available. However, the relative sparsity of the network is the reason that node-dependence (and not dyad-dependence) is sufficient for analysing IBD networks, as the dependency is not so strong as to change analyses.

To directly investigate and address this, we have added a full dyad-dependent simulation analysis to the study to show that it was not driving the performance or outcomes of the method. We do this by simulating networks with additional dyad-dependence (the amount of which was

rigorously estimated via an Approximate Bayesian computational analysis of the empirical data). We also believe that networks are unlikely to be subject to more dependence, as this study is at the pedigree level, with many first- and second-degree relatives. In doing this, we find that the correct variables are still always identified, and that overall, only ~2% of models were misclassified. However, these misclassifications are somewhat superficial, and to see this, consider that the simulated models that were misclassified were:

- Model 1 (site match) classified as Model 2 (differential site match),
- Model 2 (differential site match) classified as Model 1 (differential site match),

meaning that the only incorrect qualitative inferences made from model selection would be to assume that sites were more (or less) interconnected compared to one another than was true. We believe this shows that using only a node-dependent model is the best current option, and that results are not compromised by this limitation.

For clarity, we now state in the main text that “It must be noted that this allows us to consider only dyad-independent ERGMs, and we make this choice for the reasons of computational efficiency, due to potentially large numbers of nodes in the network, and model degeneracy due to the likely relative sparsity of the network”. We note that Reviewers 2 and 3 are supportive of the use of our dyad-independent approach, and Reviewer 3 specifically notes that this is a “sensible” limitation.

We now make an explicit statement in the text that we consider only dyad-independent ERGMs. Nevertheless, we appreciate the comments, and have added some additional content to address this (see below also).

To provide an example - suppose a family with one set of siblings of size 'n' is buried all with the same orientation - say $n=6$. Now let's double the 'n' to $n=12$ - should we take this as twice the evidence in favor of orientation matches leading to IBD matches? (Presumably not as in either case it's really just 1 family making a choice of an orientation and then a set of correlated observations). Or let's suppose we observed 3 families with $n=2$, where each pair of buried siblings was buried in the same orientation - would we take that as equal to the original? (Presumably not as the 3 families would represent perhaps 3 independent choices of orientation for the siblings rather than 1).

This is an interesting thought experiment! To test this, we simulated basic networks, although we used slightly more nodes as ERGMs simply cannot make stable estimates on such small networks. To this end, we made one network according to the first example (*family of size $n=6$*) with ten families with 6 children, of which 3 families have children buried uniquely, a second network (double the 'n' to $n=12$) ten families with 12 children, 3 of which again have the children buried uniquely, and lastly (3 families with $n=2$) 30 families with two children, 9 of which have the children buried uniquely. In each case we make one parent share the unique burial practice (this makes sure that there are connections between Normal-Normal, Normal-Unique and Unique-Unique for stability reasons). These simulations can be seen in the extra figure below.

Hence we capture your suggested dynamics: a baseline (Panel A, Sim 1), groups just like the baseline, but twice as big (Panel B, Sim 2) and groups like the baseline, but spread out over smaller sub-groups (Panel C, Sim 3). The Figure shows these networks (A-C) but also the resulting coefficient estimates from the ERGMs (D) for the “intercept” of the ERGM (edges), plus the additional log-odds for both individuals being “Normally buried” (Normal) or both being “Uniquely buried” (Unique), compared to a pair of individuals buried differently. Note that we attempted to analyse these networks with dyad-dependence terms, but found no solutions due to model degeneracy.

First, the estimates of the effects of similar burial types appears to be roughly equal for the first two simulations (D, Sim1 and Sim 2). We expect the effects of identical burial pairs to be slightly higher, as there are more connected pairs (8.54% of pairs connected in Sim 1 vs 9.24% in Sim 2). This should be a subtle adjustment, and it is, and certainly not twice the evidence as you predicted.

Second, the more spread out comparison (D, Sim1 and Sim3) shows a slightly subtle change in the opposite direction, and for similar reasons, since only 2.1% of pairs are connected here.

However, we do not see a doubling or halving of these coefficients. However, we would again expect no change in the relative proportion of related individuals sharing this practice has not changed by some significant amount.

February 17, 2026

RE: GENETICS-2025-308872

Dr. Adam B Rohrlach
Max-Planck-Institut für evolutionäre Anthropologie
Department of Archaeogenetics
Deutscher Platz 6
Leipzig 04103
Germany

Dear Dr. Rohrlach:

Congratulations, your manuscript titled "Detecting and Quantifying Networks of Biological Kinship via Exponential Family Random Graph Models" is accepted for publication in GENETICS! Many thanks for submitting your research to the journal.

The reviewer, who was previously quite critical, is happy with your revisions, and with your response to their concerns. I think that this review process has substantially strengthened the paper. The reviewer did ask, however, that you refer in the main text specifically to the additional analyses that are now in the SI - please make these changes in your final submitted version. You can view the reviewer's comments at the bottom of this email.

To Proceed to Publication:

1. Format your article according to GENETICS style: <https://academic.oup.com/genetics/pages/author-guidelines>
2. Ensure that you comply with data and community resource citation guidelines: <https://academic.oup.com/genetics/pages/author-guidelines#section-5-9-2>
3. Upload your final files at <https://genetics.msubmit.net>
4. Add oupsupport@scipris.com and genetics.oup@novatechset.com (or the domains @scipris.com and @novatechset.com) to your email program's "safe senders" list. You will be contacted by both at various points during the production process.

Notes:

- Your currently-accepted manuscript (unedited, as submitted, reviewed, and accepted) will be published at GENETICS and deposited into PubMed as an Advance Access article. Notify sourcefiles@thegsajournals.org before signing your license if you do not wish to publish your article via Advance Access.
- We invite you to submit an original color figure related to your paper for consideration as cover art. Please email your submission to the editorial office or upload it with your final files. You can submit a small-sized image for evaluation, and if selected, the final image must be a TIFF file 2513px wide by 3263px high (8.375 by 10.875 inches; resolution of 600ppi). Please avoid graphs and small type.
- After files are sent to Oxford University Press we use SciPris to manage article licensing and payment. If you do not have a SciPris account, you will receive an email from no-reply@scipris.com to sign up to use Oxford University Press' author portal. After logging in, follow the online instructions to sign your license and arrange any payment due.

If you have any questions or encounter any problems while uploading your accepted manuscript files, please email the editorial office at sourcefiles@thegsajournals.org.

Sincerely,

Nick Barton
Senior Editor
GENETICS

Approved by:
Howard Lipshitz
Editor in Chief
GENETICS

Review comments (if applicable):

Reviewer #3 :

The authors have adequately addressed my comments. Several new analyses have been added in the SI (sensitivity to the cutoff c , ABC analysis of dyadic dependance), which are crucial, but have not been directly addressed in the main text. I suggest adding direct references (appropriately contextualized) to each of these new analyses and figures in the main text itself.